# Surface water monitoring in small water bodies: potential and limits of multi-sensor Landsat time series

Andrew Ogilvie[1,2], Gilles Belaud[1], Sylvain Massuel[1], Mark Mulligan[2], Patrick Le Goulven[1], and Roger Calvez[1]

[1]G-EAU, AgroParisTech, Cirad, IRD, IRSTEA, Montpellier SupAgro, Univ Montpellier, Montpellier, France
[2]Department of Geography, King's College London, WC2R 2LS London, UK

*Correspondence to:* Andrew Ogilvie (andrew.ogilvie@ird.fr)

**Abstract.** Hydrometric monitoring of small water bodies (1-10 ha) remains rare, due to their limited size and large numbers, preventing accurate assessments of their agricultural potential or their cumulative influence in watershed hydrology. Landsat imagery has shown its potential to support mapping of small water bodies but the influence of their limited surface areas, vegetation growth and rapid flood dynamics on long term surface water monitoring remains unquantified. A semi-automated

method is developed here to assess and optimise the potential of multi-sensor Landsat time series to monitor surface water extent and mean water availability in these small water bodies. Extensive hydrometric field data (1999-2014) for 7 small reservoirs within the Merguellil catchment in Central Tunisia and SPOT imagery are used to calibrate the method and explore its limits. MNDWI is shown out of six commonly used water detection indices to provide high overall accuracy and threshold stability during high and low floods, leading to a mean surface area error below 15%. Applied to 546 Landsat 5, 7 and 8 images

over 1999-2014, the method reproduces surface water extent variations across small lakes with high skill ($R^2 = 0.9$) and mean RMSE of 9300 $\mathrm{m}^2$. Comparison with published global water data sets reveals a mean RMSE of 21 800 $\mathrm{m}^2$ (+ 134%) on the same lakes and highlights the value of a tailored MNDWI approach to improve hydrological monitoring in small lakes and reduce omission errors of flooded vegetation. The rise in relative errors due to the larger proportion and influence of mixed pixels restricts surface water monitoring below 3 ha with Landsat (NRMSE = 27%). Interferences from clouds & scan line

corrector failure on ETM+ after 2003 also decrease the number of operational images by 51%, reducing performance on lakes with rapid flood declines. Combining Landsat observations with 10 m pansharpened Sentinel-2 imagery further reduces RMSE to 5200 $\mathrm{m}^2$, displaying the increased opportunities for surface water monitoring in small water bodies after 2015.

## 1 Introduction

### 1.1 Monitoring water resources in small reservoirs

Support from governmental and international projects combined with farmer initiatives have led to significant development of small reservoirs (< 1 $\mathrm{Mm}^3$) across the globe, including India (Bouma et al., 2011; Venot and Krishnan, 2011; Massuel et al., 2014), Brazil (Burte et al., 2005) and parts of Northern and sub-Saharan Africa (Nyssen et al., 2010; Sawunyama et al., 2006;

Talineau et al., 1994). These have been built to reduce sediment transfer and silting of downstream dams, as well as harvest scarce and unreliable water resources for local users (Habi and Morsli, 2011; Wisser et al., 2010).

Despite their growing importance worldwide, small reservoirs are rarely monitored in situ, except for research purposes (Albergel and Rejeb, 1997) due to their quantity, size and geographical dispersion. Instrumenting and maintaining a hydrometric observation network requires significant time, equipment and transport costs (Liebe et al., 2005) which are not compatible with the localised and modest importance in the water budget of small reservoirs. As a result, little information is available on their water availability and their cumulative influence (Ogilvie et al., 2016b).

Water availability in small reservoirs is constrained by design capacities but determined by site-specific hydrological dynamics regulated by both natural (climate, topography, geology, geomorphology, pedology), and human factors (maintenance, leaks, withdrawals, releases). Studies based on hydrological measurements and geochemical analyses have identified the water balance of small reservoirs (Gay, 2004; Grunberger et al., 2004; Lacombe, 2007; Li and Gowing, 2005; Nyssen et al., 2010) but highlighted multiple modelling uncertainties due to evaporation, groundwater flows, high spatial rainfall variability and human management (Li and Gowing, 2005; Grunberger et al., 2004; Lacombe, 2007; Leduc et al., 2007; Kingumbi et al., 2007). Rainfall-runoff modelling of gauged small reservoirs in Tunisia notably failed to exceed $R^2 = 0.5$ (Lacombe, 2007) and difficulties are greater in ungauged catchments (Cudennec et al., 2005).

## 1.2 Remote sensing in hydrology

Images from a variety of active and passive sensors have been used successfully in hydrology to inventory, to assess flooded areas, river stage levels and widths, as well as to investigate hydrological processes and water balance issues. Low spatial resolution sensors such as MODIS (Moderate Resolution Imaging Spectroradiometer, 250 m) and AVHRR (Advanced Very High Resolution Radiometer, 1.1 km) providing daily coverage of the globe, notably facilitated the assessment and monitoring of large wetlands (Guo et al., 2017) including the Niger Inner Delta (Mahé et al., 2011; Seiler et al., 2009; Ogilvie et al., 2015; Bergé-Nguyen and Crétaux, 2015), the Okavango Delta (Wolski and Murray-Hudson, 2008; Gumbricht et al., 2004), the Tana Delta (Leauthaud et al., 2013) or the Mekong Delta (Kuenzer et al., 2015; Sakamoto et al., 2007), large rivers such as the Amazon (Martinez and Le Toan, 2007; Alsdorf et al., 2007) and large lakes notably in East Africa (Swenson and Wahr, 2009; Ouma and Tateishi, 2006) and China (Ma et al., 2007; Qi et al., 2009). These sensors have also been used for global assessments (Prigent et al., 2007; Papa et al., 2010; Klein et al., 2015) but their low spatial resolutions remain inadequate for small reservoirs, as a single MODIS pixel corresponds to an area of 6.25 ha.

High spatial resolution imagery such as SPOT (Satellite Pour l'Observation de la Terre), Quickbird and Ikonos capture images under 5 m resolution but their reduced spatial coverage (10-20 km wide for the highest resolution) make them less suitable for medium-sized catchments and their costs are prohibitive for operational long-term monitoring. Recent sensors such as Sentinel-2 capture 10 m images of the entire globe every 5 days, providing enhanced opportunities but hydrological investigations which require historical perspectives will continue to rely on previous sensors. Landsat which provides free multispectral images since the 1970s at medium geometric resolution (30 m since Landsat 4 in 1982) every 16 days (since

Landsat 7 in 1999, as previous sensors present multiple gaps) therefore continues to provide the most potential to detect and monitor small water bodies.

Satellite altimetry originally developed to monitor the ocean's surface has increasingly been exploited and adapted since the 1990s to monitor height variations of continental surface waters (Crétaux et al., 2016). Used across rivers and large lakes, recent works have sought to transpose these to smaller water bodies (from 50 ha) but highlight several limitations relating essentially to the poor density of altimetry tracks, the long along-track path lengths, and low revisit times (Baup et al., 2014; Avisse et al., 2017). Altimeters aboard Topex, Envisat, Jason-1 and Jason-2 have temporal resolutions ranging from 10 to 35 days and high vertical accuracy however their narrow swaths and long along track path lengths (1 km) restrict their application essentially to large lakes ($> 100 \text{ km}^2$, Avisse et al. (2017)). Other altimeters have improved spatial resolution and cover more of the globe but at the expense of low revisit times (368 days for Cryostat-2), removing any monitoring possibilities. The Dahiti altimetry database (Schwatke et al., 2015), employed by Busker et al. (2018), combines the tracks of numerous altimeters to optimise the sites covered, temporal resolution and the length of observations. These provide data for 168 sites in Africa (however none in Tunisia) and lakes must have a minimum 300 m diameter (circa 7 ha). The Sentinel-3a and 3b constellation provide major improvements in their along track resolution (300 m) making them potentially suitable for lakes around 4 ha but inter-tracks of 52 km mean many lakes are not covered by their trajectories (Crétaux et al., 2016). Furthermore, radar altimetry provides an estimate of the altitude of the water surface, based on the two-way travel time of the radar pulse and the known altitude of the satellite, but assessing absolute volumes requires site specific data such as stage height or topographic data (Baup et al., 2014). Avisse et al. (2017) showed that DEM data taken before lakes were built or when it was empty could be used but on larger lakes (59 ha to 379 ha) with 30 m data.

## 1.3   Landsat imagery to map and monitor small water bodies

Local studies on small lakes from 1 ha to 25 ha in West Africa (Gardelle et al., 2010; Liebe et al., 2005; Annor et al., 2009; Soti et al., 2010; Jones et al., 2017), in Southern Africa (Sawunyama et al., 2006), in South India (Mialhe et al., 2008), and in South America (Rodrigues et al., 2011) specifically investigated the potential of remote sensing methods to assess surface areas of small reservoirs. Field validation confirmed the potential of remote sensing (RS) methods as a cost-effective technique to identify water bodies but revealed problems of variable accuracy (Ran and Lu, 2012; Annor et al., 2009; Solander et al., 2016) on small lakes (under 100 ha). These studies provided a snapshot of floods in reservoirs at a given time (Liebe et al., 2005), or of seasonality (Annor et al., 2009; Jones et al., 2017) but did not explore their dynamics over time.

Smaller water bodies have also been included in worldwide inventories, done with Landsat imagery (Feng et al., 2016; Verpoorter et al., 2014; Yamazaki et al., 2015). Landsat was recently used to undertake long term monitoring of surface water bodies at global (Pekel et al., 2016) and regional scales (Mueller et al., 2016; Tulbure et al., 2016; Avisse et al., 2017). The remarkable study by Pekel et al. (2016) exploited 1 petabyte of Landsat archive imagery to study surface water dynamics in 30 m pixels globally over 1984-2015. Small water bodies were included, but Yamazaki and Trigg (2016) warns of limitations in Pekel et al. (2016) as "resolution issues prevent analysis of small water bodies". Similarly Mueller et al. (2016) focussing on Australian water bodies recognised the difficulties in the "presence of water and vegetation within the pixel" and that their

"product may not be fit for [...] small farm dams". In their own inventory of water bodies, Yamazaki et al. (2015) had also noted the omission errors on water bodies below 100 ha. These errors on small lakes are influenced by the insufficient spatial resolution but essentially by the increased presence of flooded vegetation and shallow waters which affect the reflectance signal (Sawunyama et al., 2006; Annor et al., 2009; Mueller et al., 2016; Yamazaki et al., 2015). These studies recognise the

difficulties of using Landsat imagery, but adequate ground truth data remains sparse and extensive field validation has not quantified the limits of these approaches on the smallest lakes. The feasibility of Landsat to monitor surface water dynamics and provide long term water availability assessments in small water bodies remains therefore to be quantified. (Baup et al., 2014; Pekel et al., 2016). Validation depends on higher resolution imagery or other published water maps which suffer from similar classification inaccuracies (Yamazaki et al., 2015), non negligible on these small water bodies. Pekel et al. (2016) for

example in their validation used sample pixels of seasonal water bodies within $1°$ tiles but these did not specifically account for small water bodies as shown in this paper.

Furthermore, hydrological monitoring of small water bodies imposes specific constraints on temporal resolution, due to the limited flood amplitude and the rapid flood declines, and "with the 16-day repeat cycle of Landsat [...] floods may be missed" (Yamazaki and Trigg, 2016). Clouds and shadows in optical imagery as well as the scan line corrector failure (after

31.05.2003 on Landsat 7 ETM+ sensor) are known to have a detrimental influence in global water data sets (Pekel et al., 2016; Mueller et al., 2016). These have a proportionally larger influence on small water bodies where gap-fill methods are not suited to localised rapid changes in land cover (Zeng et al., 2013; Zhu and Woodcock, 2014). The influence of Landsat's limited temporal resolution and these interferences on representing flood dynamics in small lakes remains however unknown.

## 1.4 Improving surface water monitoring in small reservoirs

The objective of this research was to assess and optimise the ability of Landsat to provide sufficient, accurate observations of long term surface water dynamics in the smallest water bodies (1-10 ha). Exceptional long term hydrological time series of 7 small reservoirs over up to 15 years are used here to develop and validate a semi-automated method using Landsat imagery, capable of reproducing surface water dynamics and long term water availability in small water bodies.

Numerous spectral water indices and classification methods exist but these are often tested and developed empirically on

larger water bodies (Feyisa et al., 2014) and their suitability in different settings must be investigated (Palmer et al., 2015). Suitable automation of water detection over several reservoirs and over several images also imposes specific constraints on the methodology, in terms of detection accuracy, threshold stability, cloud presence, and computer processing when manipulating satellite imagery (Mialhe et al., 2008; Ogilvie et al., 2015; Pekel et al., 2016). Widely used water detection indices were compared to assess their suitability to monitor floods on small reservoirs. The interferences from clouds, shadows and scan line

corrector failure were investigated and reduced here by defining optimal filter thresholds based on long term field data. Results were compared with analysis performed using available global water data sets (Pekel et al., 2016) to quantify accuracy issues and argue for the benefits of specific approaches for small water bodies. Finally, the benefits provided by Sentinel-2 imagery for surface water monitoring since 2015 were quantified.

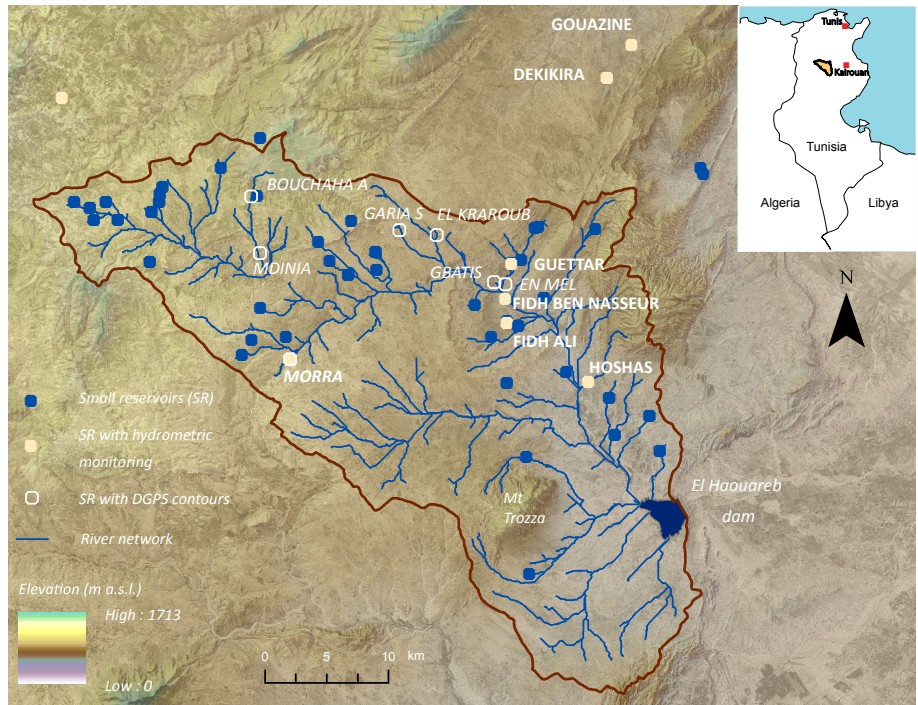

**Figure 1.** Merguellil upper catchment and location of instrumented small reservoirs

## 2 Methods

### 2.1 Small reservoirs in the Merguellil upper catchment

The case study area is the Merguellil upper catchment located in central semi-arid Tunisia (Fig. 1). A total of 58 small reservoirs were built across this 1180 km$^2$ catchment since the 1960s, as a result of international projects and an ambitious nationwide

water and soil conservation strategy. These were inventoried through the combination and cross referencing of records from local authorities, literature (Kingumbi, 2006; Lacombe, 2007; CNEA, 2006), satellite imagery and field visits (Ogilvie et al., 2016a).

Initial design capacities of small reservoirs in the catchment range between 17 000 m$^3$ and 1 590 000 m$^3$ though the median size only reaches 66 000 m$^3$. Peak flooded surface areas vary between 0.5 ha and 18 ha. Lakes flood as a result of intense

localised rainfall events (Ogilvie et al., 2016b), during autumn and spring mostly. High annual and interannual variability in rainfall (329 mm/year $\pm$131 mm) combined with significant heterogeneity in the infiltration, evaporation rates and occasional withdrawals lead to large disparities in the frequency and duration of floods in small lakes (Ogilvie, 2015; Lacombe, 2007).

Due to the importance of the downstream irrigated Kairouan plain for national agricultural production, the catchment has been the focus of numerous research projects to understand the changes and influences exerted by evolving climatic and

human dynamics (dam and small reservoir construction, irrigation withdrawals, etc.). Small reservoirs in the Merguellil upper

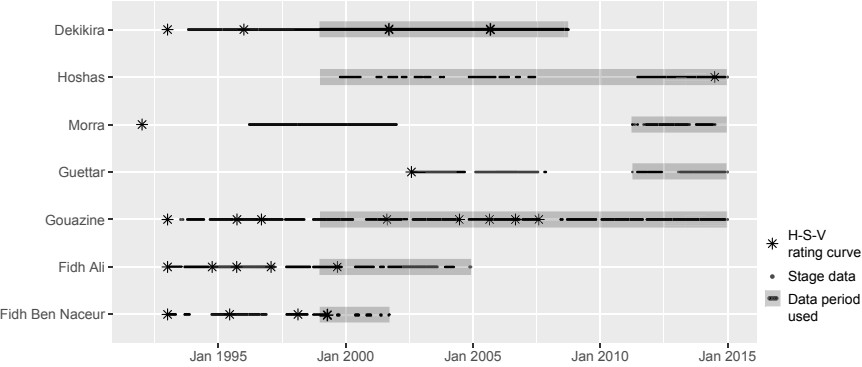

**Figure 2.** Availability of stage data and rating curves over 1992-2014 for the 7 lakes monitored

catchment and along the Tunisian NE-SW mountain range ("Dorsale") consequently benefited from extensive hydrological monitoring thanks to ongoing research collaboration with the local authorities (Albergel and Rejeb, 1997).

## 2.2 Field hydrometric monitoring

Seven reservoirs helped calibrate and validate the spectral water indices. Lakes were chosen amongst those identified as flooded on a classified 10 m SPOT image in march 2013, and flooded surface areas were measured with real time kinematic (RTK) GPS, to reduce uncertainties from stage-height rating curves. 19 GPS contours were acquired, providing a range of flooded surface areas from 0.2 ha to 7.8 ha (Table 2) to test the performance of indices.

Seven small reservoirs with extensive field data were then used to test the performance of the method over time, i.e. identify how spatial and temporal resolution affected monitoring of surface water extent dynamics and water availability. Lakes were selected based on the length and quality of their hydrometric time series (Fig. 2) and include a range of altitudes (Fig. 1), capacities (Table 3) and associated flood dynamics (amplitude and duration). With the exception of lake Morra, these 7 lakes were not used in the calibration of the indices, giving further weight and confidence to the validation of their thresholds.

Reservoir levels were monitored with automatic stage pressure transducers and daily readings by local observers on limnimetric stage ladders were used to cross check and fill gaps in automatic readings. Following the decline in the number and quality of observations exacerbated by the Tunisian revolution, complementary instrumentation was implemented on three small reservoirs: Guettar, Morra and Hoshas as part of this research. Transducer uncertainties are minimal, with a 0.2 cm resolution and theoretical ±0.25% precision.

### 2.2.1 Lake hypsometric relationships

Stage values were converted to surface areas and water volumes using rating curves available for each lake, indicated in Fig. 2. These were acquired and updated to account for silting as part of regular surveys started in the late 1990s (Albergel and Rejeb, 1997). A complementary survey was undertaken on Hoshas in 2014 with a RTK GPS and interpolated based on Delaunay

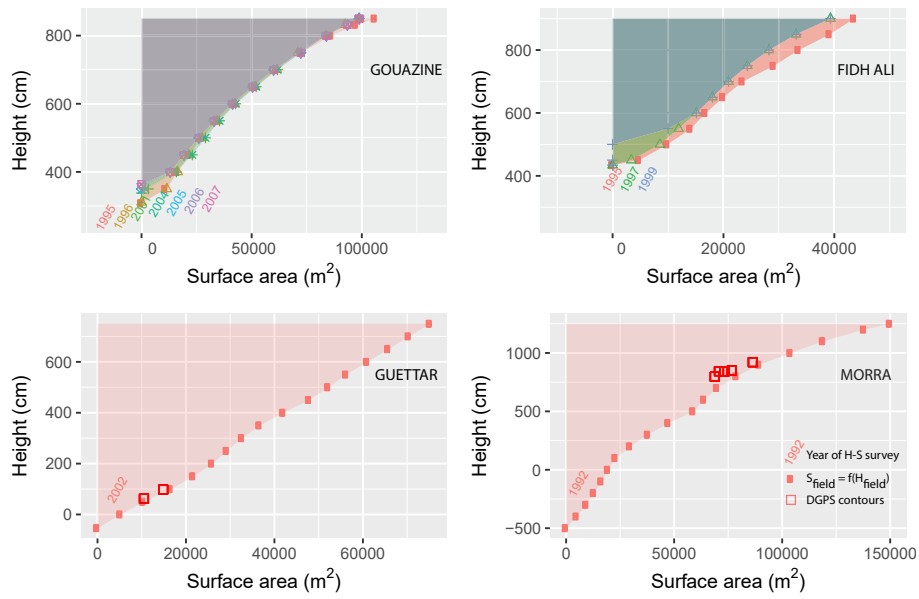

**Figure 3.** Evolution over time of stage height - surface area rating curves for 4 lakes

Triangular Interpolation Networks (TIN), well suited to bathymetric applications. Considering the logistical difficulties in updating rating curves (cost, access to lake, presence of water & dense vegetation on lakebed), complementary field data were used to assess uncertainties associated with the obsolescence of the rating curves. Figure 3 shows that mean surface area error from the rating curve against two additional GPS contours acquired on Guettar and five on Morra reaches 1450 m$^2$ and 9000

m$^2$ respectively. In relative terms these remain low (11-12%) in both cases despite rating curves not being updated for 12 and 22 years respectively. Errors rise for smaller surface areas (Fig. 3) and in catchments with higher silting rates such as Fidh Ali (Collinet and Zante, 2005; Hentati et al., 2010). Comparing surface areas calculated with the Gouazine 2001 and 2007 rating curves provided an estimate of the error generated from a 6-year old rating curve and accordingly an indication of errors on surface areas calculated until 2014. Mean RMSE reaches 2300 m$^2$ for surface areas under 10 ha but rises to 5200 m$^2$ for

surface areas below 1 ha. Similarly, on Fidh Ali after 4 years, mean RMSE reaches 4600 m$^2$ on flooded areas under 4 ha but rises for both small (6700 m$^2$ for $S < 1$ha) and larger (5600 m$^2$ for $S > 3$ha) surface areas.

Analysis in the region (Albergel et al., 2003; Lacombe, 2007) of 70 height ($H$) - surface ($S$) - volume ($V$) rating curves expressed as power relations (e.g. $V = B * S^\beta$) showed that the shift in the parameters ($B$, $\beta$) over time due to silting could be modelled through linear regression. Parameters evolve to reflect the decreasingly concave nature of the lake banks. In ungauged

reservoirs, such regional power relations are required, but relative errors can exceed 50% (Liebe et al., 2005; Sawunyama et al., 2006; Ogilvie et al., 2016a). In gauged reservoirs, these estimates also lead to rising uncertainties over time, as silting is not a linear, incremental process. Sediment transport is heterogeneous and occurs through sudden, discrete events, difficult to model in these small catchments (Hentati et al., 2010). On Morra, estimating silt deposit after 22 years leads to RMSE over 60 000

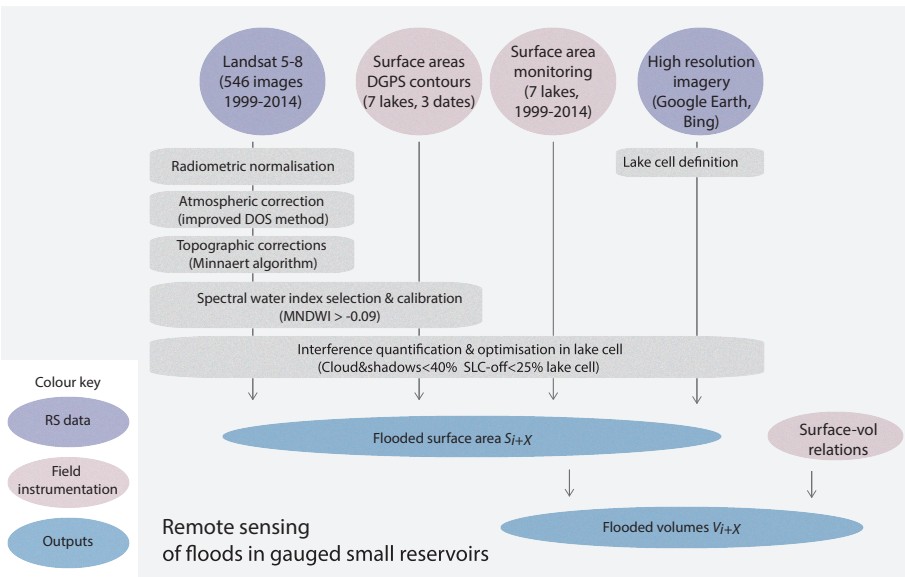

**Figure 4.** Chain of treatments applied to the 546 Landsat (TM, ETM+ and OLI) images

$m^2$ when compared to the GPS contours, confirming the difficulty of updating the initial rating curve. The influence of the rating curves on the assessment of the performance of Landsat monitoring are further discussed in the results.

## 2.3 Landsat preprocessing

Landsat Level 1 Terrain Corrected (L1T) (orthorectified) products from USGS Earth Explorer were used here. The required
radiometric (including atmospheric & topographic) corrections derived for each sensor and band combination are detailed in
the Appendix. Based on data availability of Landsat images and field limnimetric data for the seven lakes, 546 Landsat 5-8
images over the period 1999-2014 were acquired and treated (Fig. 4). Lakes cells to automate extraction of the flooded area
were defined using freely available high resolution imagery from Digital Globe and Astrium available in Google Earth and
Bing Maps. These cells defined the maximal flooded area based on visual interpretation of geomorphological boundaries of
10 the lake (with a 10% buffer) and helped constrain overestimations, i.e. pixels which do not belong to the lake but share similar
reflectance profile, such as surrounding flooded vegetation, river stretches and temporary ponds or waterlogging.

Level 2A products corrected to land surface reflectance using the Landsat Ecosystem Disturbance Adaptive Processing
System (LEDAPS) for sensors TM and ETM+ or Landsat Surface Reflectance Code (LaSRC) for the OLI sensor were not
available for our catchment and time period at the time of research. Two Landsat 8 LaSRC products now available from ESPA
were acquired for 2013 to analyse the possibility of applying the water detection approach developed in this paper directly to
level 2A imagery. Remotely sensed flooded areas were quantified and compared with 13 GPS contours to assess relative errors.

**Table 1.** Spectral water indices compared in this paper

| | |
|---|---|
| Normalised Difference Water Index (McFeeters, 1996) | $NDWI = \frac{Green - NIR}{Green + NIR}$ |
| Modified NDWI (Xu, 2006) | $MNDWI = \frac{Green - SWIR}{Green + SWIR}$ |
| Normalised Difference Pond Index (Lacaux et al., 2007) | $NDPI = \frac{SWIR - Green}{SWIR + Green}$ |
| Normalised Difference Moisture Index (Gao, 1996) | $NDMI = \frac{NIR - SWIR}{NIR + SWIR}$ |
| Normalised Difference Turbidity Index (Lacaux et al., 2007) | $NDTI = \frac{Red - Green}{Red + Green}$ |
| Normalised Difference Vegetation Index (Rouse et al., 1973) | $NDVI = \frac{NIR - Red}{NIR + Red}$ |
| Automated Water Extraction Index (Feyisa et al., 2014) | $AWEI_{nsh} = 4 \times (Green - SWIR_1)$ $- (0.25 \times NIR_4 + 2.75 \times SWIR_2)$ |

## 2.4   Spectral water indices for multi-image analysis of small water bodies

Six widely used water detection indices (Table 1) were compared to assess their suitability to monitor floods on small reservoirs in terms of detection accuracy and threshold stability. These include NDWI (Normalised Difference Water Index) developed by McFeeters (1996) exploiting the difference in reflectance of water in the green and NIR bands. Xu (2006) proposed a

modified NDWI (MNDWI) where the NIR band was substituted by the SWIR band to improve distinction of built up features over water. In parallel, Lacaux et al. (2007) developed the NDPI (Normalised Difference Pond Index) which also exploits the low reflectance of water in SWIR and its contrast with the gree band. NDPI is effectively the opposite of MNDWI and was therefore not investigated separately. Gao (1996) developed a NDWI but using NIR and SWIR bands. Xu (2006) later defined this as NDMI (Normalised Difference Moisture Index). NDTI (Normalised Difference Turbidity Index) was also developed

using the red and green bands, and exploits the principle that turbid water reacts like bare soils, i.e. low reflectance in green and high in red. The NDVI (Normalised Difference Vegetation Index) (Rouse et al. 1973) is one of the most well known band ratio index which exploits the contrast between the peak reflectance in the infrared band and the low reflectance in the red to monitor vegetation. It has also been used to detect water bodies (Ma et al. 2007; Mohamed et al. 2004) as the index becomes positive in presence of vegetation and negative for water bodies. Finally, AWEI (Automated Water Extraction Index) (Feyisa

et al. 2014) was empirically developed to discriminate water over several large lakes using wavelengths within 5 Landsat bands. Two variants exist, AWEInsh and AWEIsh, the latter being optimised to remove (urban and topographic) shadow pixels. Both can be used in succession in the presence of highly reflective areas (snow, roofs, etc.) but in this rural, semi-arid gently sloping watershed, the AWEInsh was used. For SWIR, band 5 for TM/ETM+ and band 7 for OLI were used in accordance with other studies (Ji et al. 2009; Ouma and Tateishi 2006). For NVDI and NDTI, values below the threshold were classified as water, and

conversely for the four other indices.

## 2.5 Calibration and comparison of spectral water indices

The six water detection indices were calibrated on two Landsat 8 images to assess the performance and stability of each index in high waters (March 2013), and in low waters (June 2013) when greater proportions of vegetation growth, algae, turbidity can affect water detection. A third image from May 2013 when vegetation and algae growth was evident but in reduced proportions was used to validate thresholds. Rainfall recorded over March-June 2013 ranged between 27 mm and 44 mm across the lakes concentrated on 1 event on the 24.04.2013. Timed to coincide with the 16-day Landsat 8 acquisition cycle (Table 2), the lag between satellite overpass and field observation was inferior to 1 week, similar or inferior to previous studies (Liebe et al., 2005; Annor et al., 2009; Sawunyama et al., 2006).

Georeferenced GPS contours for each lake and date (Fig. 9 and Table 2) were used to calculate confusion matrices for each index, lake and Landsat image. Classification thresholds for each index were varied in increments of 0.01, considering the observed influence of a second decimal in the thresholds (Ogilvie et al., 2015), and optimal thresholds were determined based on minimising overall accuracy rates. The median threshold from all calibration points (i.e. 7 lakes on 2 images) was retained as the optimal threshold for each index. The median was preferred over mean threshold to reduce the influence of single lakes (and dates) where the optimal threshold is significantly different due to mixed reflectance issues. Similarly, an optimal inter-lake and inter-date threshold was not determined based on aggregated pixels from all lakes, as we sought to give equal weighting to each lake rather than allowing large lakes to steer threshold values. This also ensured mixed pixels found across small reservoirs were better accounted for in the calibration (Jain, 2010). Spectral unmixing techniques (Van Der Meer, 1995) were not used here considering the marginal benefit (5% increase in sub-pixel accuracy) observed in Sun et al. (2017) and the requirements to adapt the approach to deal with residual clouds and SLC gaps in multi-temporal Landsat time series.

$$PDAI = \frac{|S_{RS} - S_{GPS}|}{S_{GPS}} * 100 \tag{1}$$

The Percentage Deviation Area Index (PDAI, Eq. (1)) (Sawunyama et al., 2006) was calculated to provide a relative measure of the spectral indices' accuracy in estimating flooded surface area remotely ($S_{RS}$) compared to the field GPS measurements ($S_{GPS}$). PDAI was not used for calibration as it is not a pixel-based assessment and was here shown to disguise spatial errors, i.e. over-classified pixels compensating for under-classified areas. Feyisa et al. (2014) showed that additional statistics notably the Kappa coefficient and McNemar's test which focus on the frequency of pixels correctly and incorrectly classified did not provide added benefit when optimising water detection thresholds (Fisher and Danaher, 2013). Confusion matrices and PDAI errors per lake on the third validation image were then used to discuss the performance of each index and determine the optimal water detection index for these small water bodies.

## 2.6 Influence of clouds, shadows and ETM+ scan-line corrector failure

Gap-fill methods designed to address the loss of pixels from SLC failure and interferences from clouds & shadows rely on nearby spatial or temporal spectral information. Suited for larger lakes and land use studies, spatial interpolation algorithms

and decision tree classifications (Maxwell, 2004; Bédard et al., 2008) assume that nearby pixels share similar reflectance profile which is not compatible with small scale analysis developed here (Zeng et al., 2013; Feng et al., 2016). Our objects of interest vary between 1 ha and 30 ha and are easily contained within a single SLC-off band, or individual clouds. SLC failure led to the loss of around 22% of pixels (Zeng et al., 2013), distributed along oblique lines ranging from 14 pixels wide at the edges to 2 at the centre (i.e. 420 m to 60 m) (Maxwell, 2004).

Temporal methods (Zeng et al., 2013; Zhu and Woodcock, 2014, 2012; Scaramuzza et al., 2004) as those employed by the USGS to provide SLC corrected images exploit additional spectral data from another image close in time. These rely on the relative stability of pixel reflectance between images of nearby dates or at least assume "similar patterns of spectral differences between dates" (Chen et al., 2011; Zeng et al., 2013). These are again not suited to the detection of ephemeral localised changes (Zhu and Woodcock, 2014; Yin et al., 2016), such as small scale, rapid changes in water surface area, which do not follow the same gradual pattern as nearby land use (vegetation) between successive images.

The Fmask algorithm (Zhu and Woodcock, 2012) was used here to quantify the presence of clouds and shadows over each lake on each Landsat image (details in Appendix). Level 1 Quality Assessment bands provided by USGS now employ Cmask, a C version of this cloud detection algorithm, therefore providing comparable data (Foga et al., 2017). Errors from clouds, shadows and SLC-off pixels were then assessed and minimised in post-processing by removing images for each lake with interferences above a determined threshold. This optimal threshold was determined by minimising the root mean squared error (RMSE) on surface area aggregated over time (15 years) in order to give importance to both the number and quality of the Landsat observations over time, and therefore compromise between maintaining high temporal resolution and minimising spatial errors from clouds, shadows and SLC-off interferences. Thresholds were varied in 5% increments and the quality of fit in terms of $R^2$ and RMSE between remotely sensed surface area and field surface area was calculated for decreasing subsets of images. Gouazine lake (Nasri et al., 2004; Al Ali et al., 2008) which possessed both the longest time series (over 15 years) and the most accurate data and rating curves (updated 6 times) was used to optimise the thresholds. The approach was repeated on four other lakes (Morra, Dekikira, Guettar, Fidh Ali) to confirm the suitability of the thresholds and explore the resulting availability of suitable Landsat imagery over all lakes.

## 2.7 Remote monitoring of surface water dynamics and water availability

The complete method was applied to all 546 images and surface water extents were assessed for the 7 small reservoirs over 1999-2014. Values affected by excessive cloud & shadow & SLC presence over each lake were removed and remaining observations were linear interpolated over time into continuous time series of flood dynamics. Alternate smoothing and interpolation of discrete observations were shown not to improve the results (Ogilvie, 2015), considering the abrupt flood dynamics in small water bodies. These were then converted to volumes using the site-specific updated surface-volume rating curves. The mean water availability per year was calculated to assess the suitability of the approach to derive long term water availability data for stakeholders. Outliers were further constrained by the known initial maximum capacity of lakes. The skill of the method in reproducing surface water extent dynamics and water availability assessments over the seven reservoirs were discussed in light of $R^2$ and RMSE values against the extensive field based data.

## 2.8 Comparison with published global water data sets

Results were compared to those obtained when using the global surface water data sets produced by Pekel et al. (2016) with 30 m Landsat imagery. These exploited original approaches based on a combination of expert systems, visual analytics and evidential reasoning to classify pixels as water, land or NA. 200 monthly water history rasters produced by the European Commission Joint Research Centre (JRC) for our region of interest were downloaded through Google Earth Engines. Monthly surface water extents areas over 1999-2014 were extracted from the raster files for the 13 lakes used previously. Confusion matrices between three JRC images and the DGPS contours from 2013 on 7 lakes were first calculated to provide pixel-based comparisons of the method's ability to detect water in the smallest reservoirs. Additionally, high spatial resolution (10 m) SPOT 5 multispectral imagery from 26.05.2013 was used to identify standing and floating vegetation on the 7 lakes based on superior NDVI (Normalised Difference Vegetation Index) values, and discuss the performance of the expert systems approach in pure and vegetated pixels. Monthly surface water values extracted from JRC datasets over 1999-2014 for 7 small reservoirs were then compared with mean monthly surface areas based on in situ monitoring. RMSE and NSE values on monthly surface water dynamics as well as mean annual water availability were again calculated to compare with results from our approach. In parallel, the number of NA values in monthly water history rasters within each lake grid cell were collected for each image to reduce their detrimental influence on surface water area estimates. NA values result from clouds and SLC-off interferences which are not differentiated in the JRC global water data sets.

## 2.9 Comparison with Sentinel-2 imagery

Sentinel-2A launched in June 2015 by the European Space Agency provides freely available multispectral imagery at 10 m spatial resolution (20 m in SWIR band) every 10 days. Its constellation with Sentinel-2B launched in March 2017 increases temporal resolution to 5 days. 84 surface reflectance products over 2015-2017 distributed by THEIA were used to quantify the performance of Sentinel-2 imagery and its combination with Landsat imagery after 2015 in terms of RMSE and NSE for daily surface water monitoring. 50 additional Landsat 7 and Landsat 8 images were processed to surface reflectance through our treatment chain to extract flooded areas and additional field data from ongoing in situ monitoring on 1 lake (Gouazine) was acquired.

THEIA Sentinel-2 products include atmospheric corrections based on the MAJA (MACSS-ATCOR Joint Algorithm) treatment chain developed by CNES-CESBIO (Hagolle et al., 2015) and German DLR. These notably include 30 m topographic corrections, as did the Landsat imagery. Intensity-Hue-Saturation (IHS) pansharpening (Carper et al., 1990) was used on SWIR bands to allow water detection at 10 m (Du et al., 2016; Kaplan and Avdan, 2018) with MNDWI and compare performance with 20 m and 30 m MNDWI. The optimal MNDWI threshold on Sentinel-2 imagery was independently calibrated against k-means (unsupervised) classification (Jain, 2010) of flooded areas, and was substantially lower (-0.2) than with Landsat imagery (-0.09). Cloud masks provided by the MACCS algorithm were used directly, to reduce difficulties with Fmask and Sen2cor methods resulting from the absence of the thermal band.

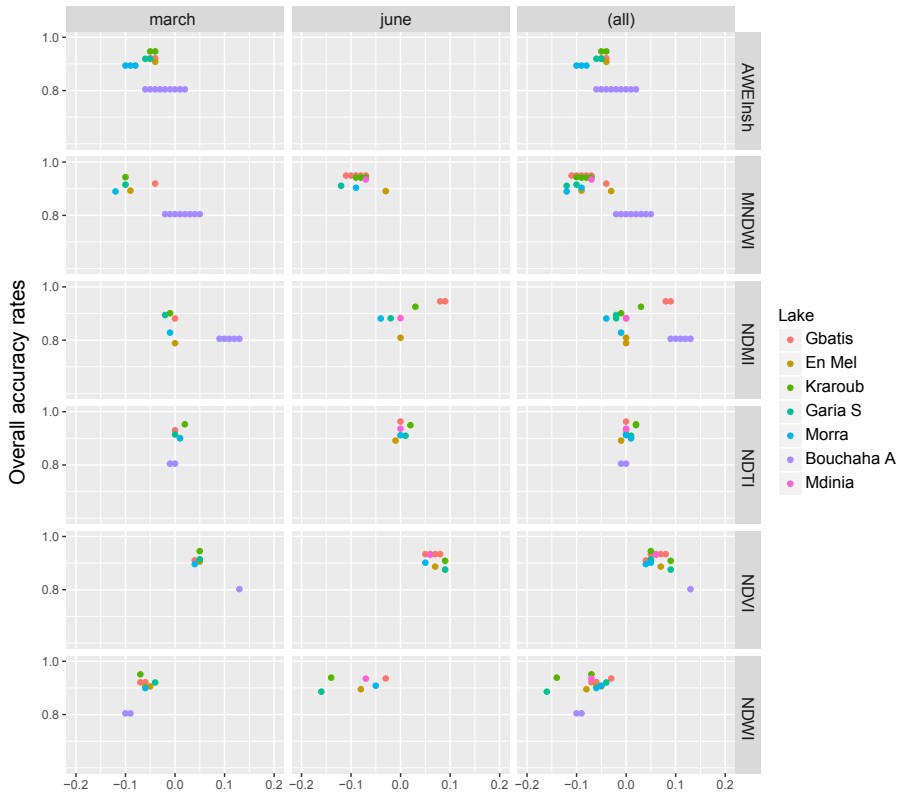

**Figure 5.** Optimal threshold (based on overall accuracy) for each lake & index on March and June calibration images

## 3 Results & discussion

### 3.1 Spatial accuracy of spectral water indices in small water bodies

#### 3.1.1 Water classification with variable thresholds (calibration)

All six spectral water indices performed well when calibrated for a single image (March or June), achieving consistently high

5 overall accuracy rates (Fig. 5) over 80%. In high waters (March), associated surface area errors remain below 20% on 4 out of 6 lakes. Even on the smallest lake (Gbatis) the PDAI values varied between 9% and 27% depending on the index, showing remarkable precision considering its size of 1 ha and the coarse 30 m image resolution.

In low waters (June), PDAI values degraded however for NDVI, NDWI and NDMI and exceeded 50% on the smallest lake. Only MNDWI and NDTI maintained PDAI rates below 25% for all lakes. In the dry season, lower water levels, greater

10 algae and vegetation growth make accurate discrimination more complex (Ji et al., 2009). Smaller lakes intrinsically have a higher proportion of mixed pixels and shallow waters (Mialhe et al., 2008) where the spectral reflectance is a combination of

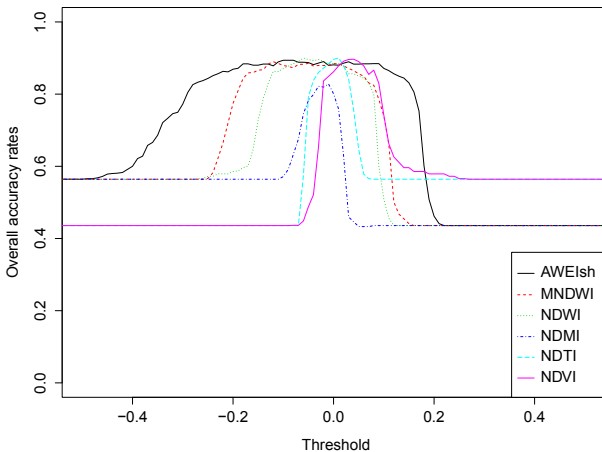

**Figure 6.** Rate of change of overall accuracy rates for varying thresholds of MNDWI on Landsat 8 29.03.2013 image for lake Morra

water, soil and vegetation. The coarse resolution exacerbates these errors due to the proportionally greater influence of a single misclassified pixel on the result (1 ha corresponds to a little more than 9 pixels).

NDVI which suffers from inability to correctly distinguish vegetation from flooded vegetation sees its performance diminish accordingly. NDMI on the other hand performed better in low waters, coherent with its objective to detect moisture and all
5 types of water. Thanks to the middle infrared band which penetrates water less and is less affected by turbidity & vegetation in the water, NDMI was able to detect the two smallest lakes but low user accuracy led to gross PDAI errors, as it overclassified many vegetated pixels. MNDWI and NDTI fared the best with these smaller reservoirs (eg. Gbatis) due to their ability to detect turbid waters, and distinguish flooded vegetation over wet vegetation on land.

AWEI performed well on the first image, but the index failed on the other images with overall accuracies falling below 60%
on four lakes. These were here related to inherent discrepancies in the radiometric corrections apparently amplified by the nature of the AWEI equation, rather than contained as with other normalised band ratio indices. Normalising reflectance values between the 546 images may help reduce these issues, but as a result AWEI was not considered for the semi-automation of this method. Furthermore, Jiang et al. (2014) showed that MNDWI outperformed AWEI when extracting narrow water bodies on Landsat.

### 3.1.2 Water classification with fixed thresholds (validation)

The optimal threshold determined from the March and June images on seven lakes was used in validation on a third image (May). Accuracy rates remained high for MNDWI and NDTI but reduced significantly for the three other indices (Fig. 7). Issues of shallow waters and vegetation were again prevalent here and even an image-specific threshold could only minimise omission and commission errors to 50% with NDVI, resulting in PDAI of 46%.

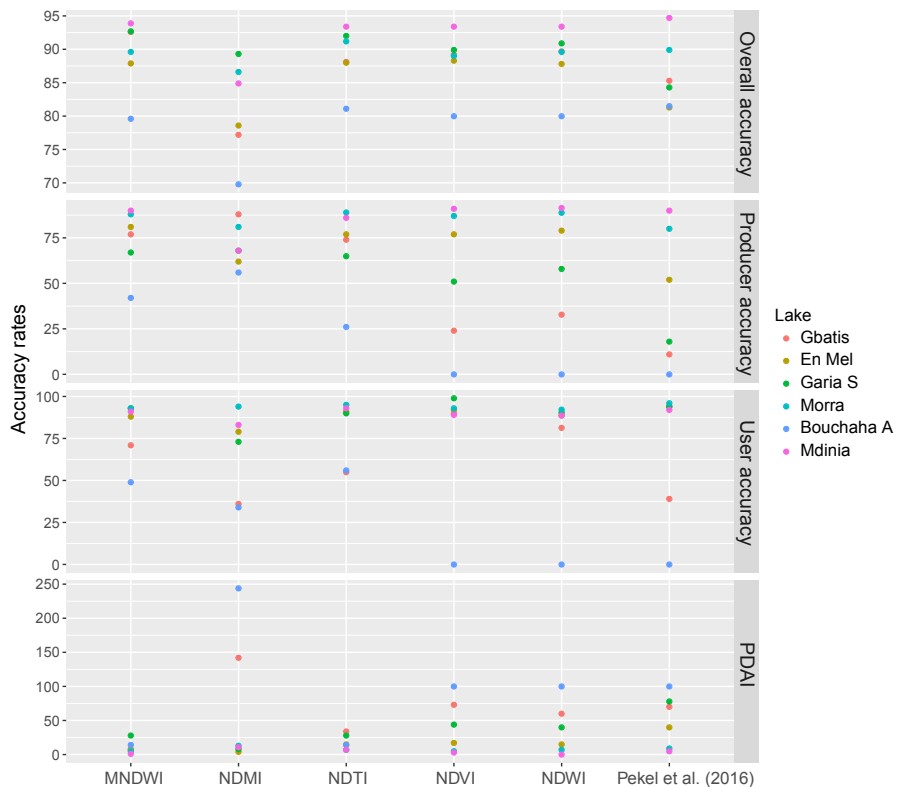

**Figure 7.** Accuracy and error values for 5 spectral water indices and the Pekel et al. (2016) expert systems approach on May validation image

The stability of the individual thresholds (Fig. 5) combines with the sensitivity to threshold change (Fig. 6) to determine the amplitude of the errors when using fixed thresholds. A marginal shift in threshold due to shallow waters and abundant algae will then introduce larger errors and omission rates with NDVI and NDWI reached 76% and 100% on the smallest lakes. The significant rate of change observed highlighted that thresholds must be optimised up to 2 decimals. MNDWI achieved better
5   accuracies on these small lakes and achieved consistently the lowest PDAI errors, as a result of the lower rate of change of the MNDWI index over NDTI (Fig. 6).

These results confirm firstly its suitability over other band ratio indices on the smallest lakes, in line with results on larger water bodies (Feyisa et al., 2014; Ji et al., 2015; Jiang et al., 2014; Ogilvie et al., 2015). Applied across seven lakes and three images, the method yields a $R^2$ Pearson correlation coefficient of 0.99 and low RMSE errors, confirming secondly that a fixed-
10   threshold MNDWI can be used to study over time lakes from 1 ha with 30 m Landsat images (Table 2). Surface area errors ranged from 1 to 28% (mean 10.5%, s.d. 8.7%) with some underestimations observed due to undetected narrow inlets on certain lakes. On the smallest lakes (Bouchaha A and Gbatis) however errors reached only 14 and 8%. In the dry season, Bouchaha A reduced to only 0.2 ha, therefore PDAI error reached 50% with all indices which is to be expected as it corresponds to just over 2 Landsat 30 m pixels.

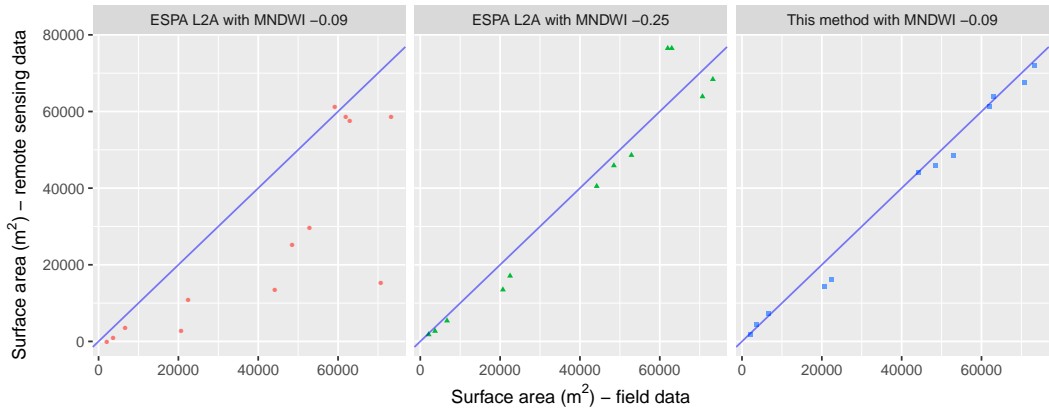

**Figure 8.** Remotely sensed surface areas with MNDWI on Landsat ESPA surface reflectance products and Landsat surface reflectance through our treatment chain versus field data (13 DGPS contours) in May and June 2013

Optimising thresholds for each lake and image individually can reduce PDAI errors below 5%, highlighting the potential for further improvements by developing (unsupervised) classification methods capable of being used over numerous locations and images reliably and efficiently (in terms of CPU and user input). Optimising thresholds for each lake but not each image only improved performance on certain small lakes.

Comparison between the Landsat surface reflectance products (level 2A) provided by ESPA and 13 DGPS contours revealed that lake surface areas are significantly underestimated on all lakes, except Mdinia (Fig. 8) when using the MNDWI threshold determined here (-0.09). RMSE reaches 22 000 $m^2$ with the L2A products compared to 3200 $m^2$ with surface reflectance products through our treatment chain, largely resulting from underestimation in small lakes. Lowering the MNDWI threshold further (to -0.25) to increase the classification of water with standing vegetation can reduce RMSE to 7000 $m^2$ but leads to

overestimations (over 20%) on large lakes such as Mdinia. Considering the good performance of the atmospheric corrections provided by ESPA's 6S Landsat Surface Reflectance Code (6S-LaSRC) (Doxani et al., 2018; Vermote et al., 2016), the greater dispersion in the reflectance values observed may be explained by the finer topographic corrections performed in our treatment chain (30 m SRTM vs. 1 km GCM DEM by LaSRC), necessary when monitoring small water bodies.

### 3.1.3   Comparison with previous approaches

Previous studies to map small reservoirs yielded PDAI errors of 10% to 50% on lakes in the range 1.5 - 5 ha and with image-specific thresholds (Liebe et al., 2005; Sawunyama et al., 2006). Errors in this study remained minimal even on lakes of this size despite the same spatial resolution of the images, pointing to the improved ability of indices in detecting mixed pixels caused by shallow waters. Including mixed pixels from small reservoirs in the calibration also improved performance (Ji et al., 2009; Jain, 2010) and led to a slightly lower threshold (-0.09). The MNDWI threshold for water detection is often near 0 but

here a value close to that found in wetlands (Ogilvie et al., 2015) with similar mixed spectral profiles was determined.

**Table 2.** Landsat 8 MNDWI flooded surface area assessments compared to field measurements for 7 lakes on the calibration and validation images

| | Lake name | Gbatis | En Mel | Kraroub | Garia S | Morra | Bouchaha A | Mdinia |
|---|---|---|---|---|---|---|---|---|
| | Cell n° | 18 | 21 | 25 | 28 | 30 | 35 | 50 |
| Calibration | L8 surface area 29.03.13 (ha) | 1.08 | 5.94 | 8.01 | 2.16 | 8.19 | 0.18 | 7.56 |
| | GPS surface area 26.03.13 (ha) | 0.99 | 5.60 | 7.78 | 2.51 | 7.67 | 0.18 | NA |
| | Overall accuracy (%) | 91.4 | 89.3 | 94.1 | 89.7 | 87.7 | 76.0 | NA |
| | Producer accuracy (%) | 79 | 89 | 88 | 63 | 90 | 40 | NA |
| | User accuracy (%) | 79 | 86 | 88 | 84 | 89 | 45 | NA |
| | PDAI (%) | 9 | 6 | 3 | 14 | 7 | 0 | NA |
| | L8 surface area 09.06.13 (ha) | 0.45 | 4.41 | 4.86 | 1.44 | 6.75 | 0.09 | 6.39 |
| | GPS surface area 11-14.06.13 (ha) | 0.37 | 4.42 | 5.29 | 2.07 | 7.07 | NA | 6.30 |
| | Overall accuracy (%) | 94.9 | 88.2 | 94.2 | 90.0 | 90.4 | NA | 92.6 |
| | Producer accuracy (%) | 78 | 80 | 77 | 62 | 88 | NA | 87 |
| | User accuracy (%) | 63 | 87 | 87 | 90 | 93 | NA | 90 |
| | PDAI (%) | 22 | 0 | 8 | 30 | 5 | NA | 1 |
| Validation | L8 surface area 24.05.13 (ha) | 0.72 | 4.59 | 17.37* | 1.62 | 7.20 | 0.18 | 6.12 |
| | GPS surface area 21.05.13 (ha) | 0.67 | 4.85 | 5.92 | 2.25 | 7.33 | 0.21 | 6.20 |
| | Overall accuracy (%) | 92.6 | 87.9 | 64.8 | 92.7 | 89.6 | 79.6 | 93.9 |
| | Producer accuracy (%) | 77 | 81 | 86 | 67 | 88 | 42 | 90 |
| | User accuracy (%) | 71 | 88 | 38 | 93 | 93 | 49 | 91 |
| | PDAI (%) | 8 | 7 | 244 | 28 | 5 | 14 | 1 |

* 46% of clouds and 50% shadows were detected over this lake on this L8 image

Confusion matrices performed on the JRC monthly surface water rasters (Pekel et al., 2016) confirmed their excellent performance on larger lakes in high waters (e.g. Morra, Mdinia) with PDAI under 10% and overall accuracies near 90%. Overall accuracy values remained high (> 80%) on other lakes, due to the large number of non water pixels correctly classified in each lake cell but water omission errors (i.e. producer accuracy) on the smaller lakes (under 5 ha) were substantial, rising in some cases over 50% (Fig. 7). Commission errors remained minimal (10% on average) indicating that the expert system approach suffered from the inability to detect all types of water, rather than from overdetection. Confusion matrices on the June images confirmed the progressive decline in accuracy as water reduces and the presence of flooded vegetation, algae and shallow waters increase. In June, overall accuracy rates declined and PDAI for instance rose from 9.1% to 18.7% on Morra and up to 74.7 % on Kraroub where significant algae and vegetation growth was observed. In comparison, MNDWI designed to detect mixed water reflectance led to a 7.1 % PDAI on this 5 ha lake. Across all 7 lakes, omission errors in JRC datasets

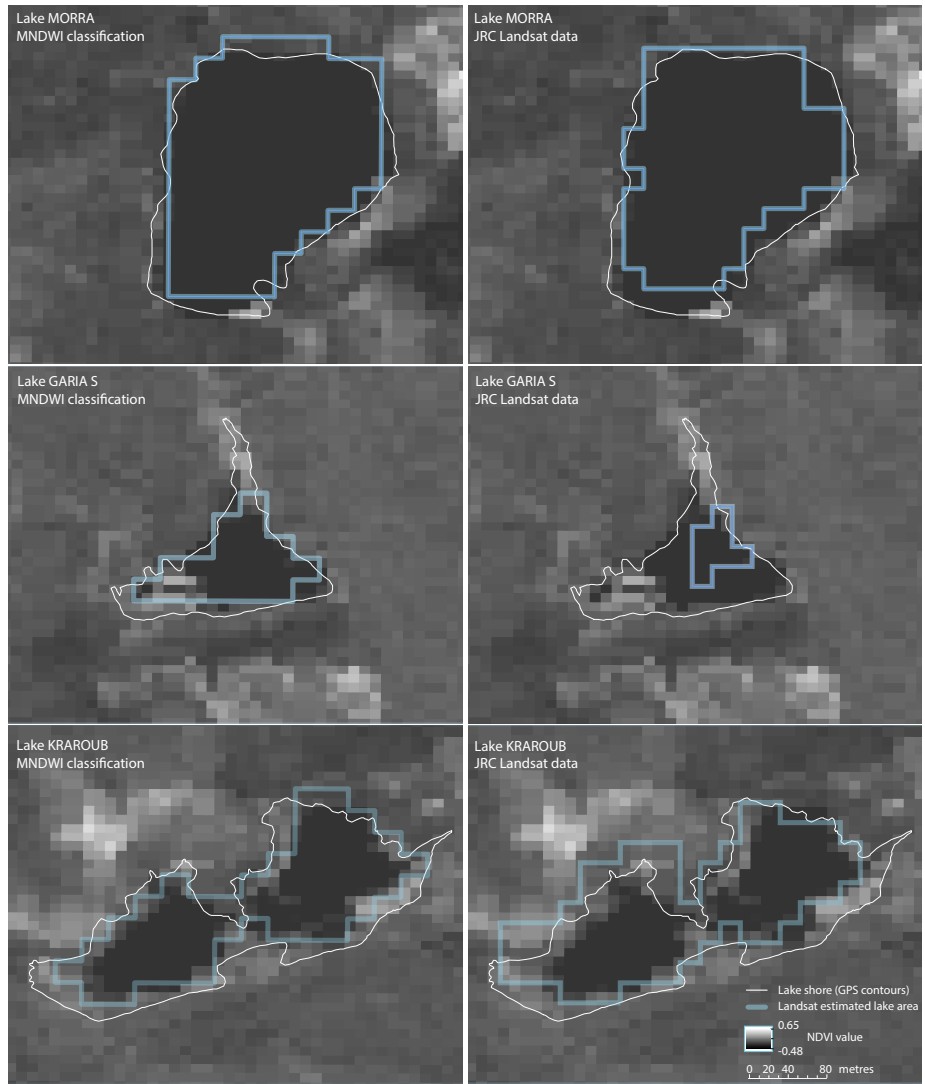

**Figure 9.** Comparison of surface water classification from MNDWI approach and JRC data against DGPS contours and SPOT 5 NDVI maps for 3 lakes in May 2013

reached 32% in high waters and 41% in shallow waters. These are substantially higher than values provided in Pekel et al. (2016) (23% for seasonal water bodies in Landsat 8) and in Mueller et al. (2016) (12% in small water bodies and 26% where water and vegetation mix). These suggest that the validation approach which uses sample pixels with "high probability of water occurence" within $1°$ tiles (Pekel et al., 2016) may introduce a bias towards larger, near permanent water bodies, which also contained greater proportions of clear water pixels.

NDVI maps created from high resolution (10 m) SPOT imagery confirmed the accurate detection of pure water pixels by both methods and illustrate the difficulties with the JRC datasets where lakes include standing or floating vegetation (Fig. 9).

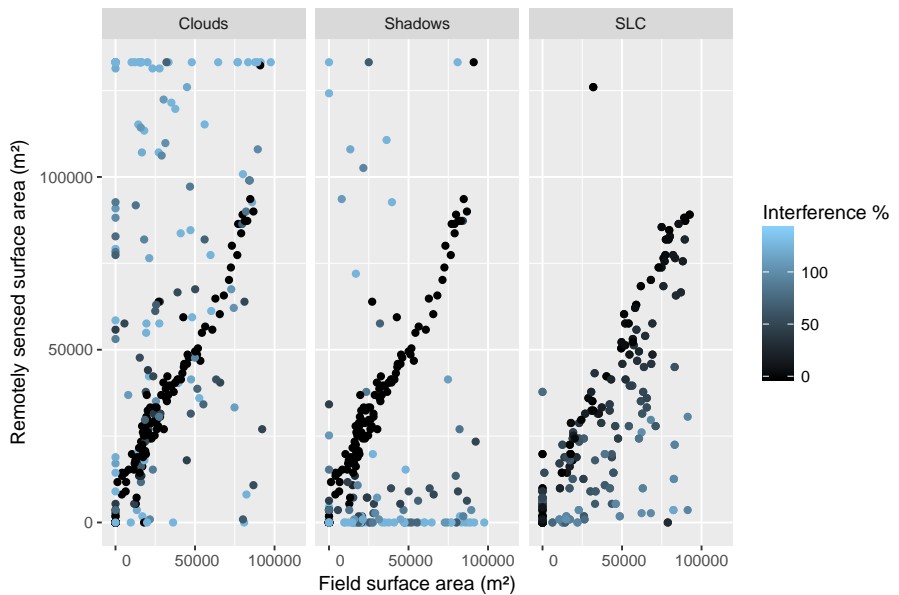

**Figure 10.** Errors due to interference from clouds, shadows and SLC-off percentage across Gouazine lake cell

The lake fringes are notably excluded by the JRC approach, which on larger lakes such as Morra and Mdinia leads to minor underestimation of the flooded area. However, on lake Garia S, where vegetation is observed on much of the west of the lake, this leads to substantial underestimations (PDAI = 80%). NDVI maps also highlight additional pure water pixels in the south of Garia S and En Mel lakes which are excluded by the expert systems approach. Conversely, on lake Kraroub, the JRC datasets

included vegetated pixels in May 2013 but not in June 2013. Though results point largely to difficulties in detecting shallow waters where reflectance is affected by vegetation and turbidity, other factors influencing the JRC approach such as terrain shadows and lake morphology could not be excluded here. The good performance on Kraroub in May 2013 confirms that in some situations, the expert systems approach is capable of reaching its goal of detecting water even with variable "chlorophyll concentration, total suspended solids and coloured dissolved organic matter load" (Pekel et al., 2016), but not systematically.

### 3.2 Influence of clouds, shadows & SLC-off pixels on multi-temporal analysis

#### 3.2.1 Interference from clouds, shadows & SLC-off

The influence of clouds, shadows and SLC failure on estimated flooded areas from Landsat imagery over 1999-2014 are illustrated in Fig. 10 and Fig. 11 based on field data from one small reservoir. Detected on 28% of the 546 Landsat images, clouds lead essentially to commission errors (false positives) and overestimation, as a result of visible wavelengths being

reflected while much of the electromagnetic energy is absorbed by the droplets in clouds. Their effect was not systematic and proportional (Fig. 11), as shown by underestimations in 30% of cases where clouds were detected across the whole lake cell. The diversity in the nature and properties (temperature, thickness, water content...) of clouds are indeed firstly responsible for

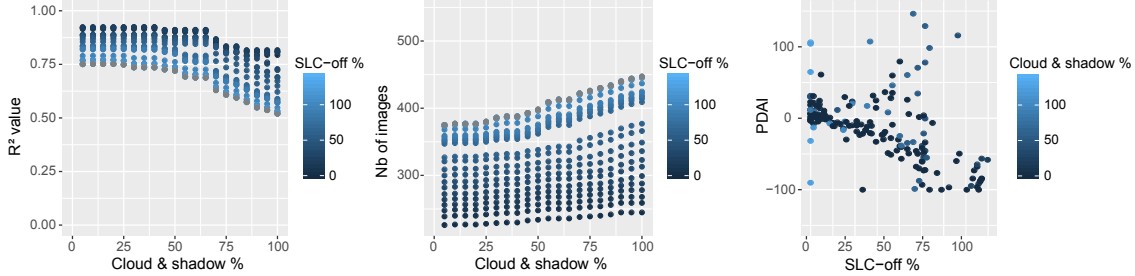

**Figure 11.** Change in $R^2$ (between field and remotely sensed surface areas), remaining number of Landsat scenes and PDAI for increasing percentages of SLC-off and cloud & shadows pixels tolerated (Lake Gouazine)

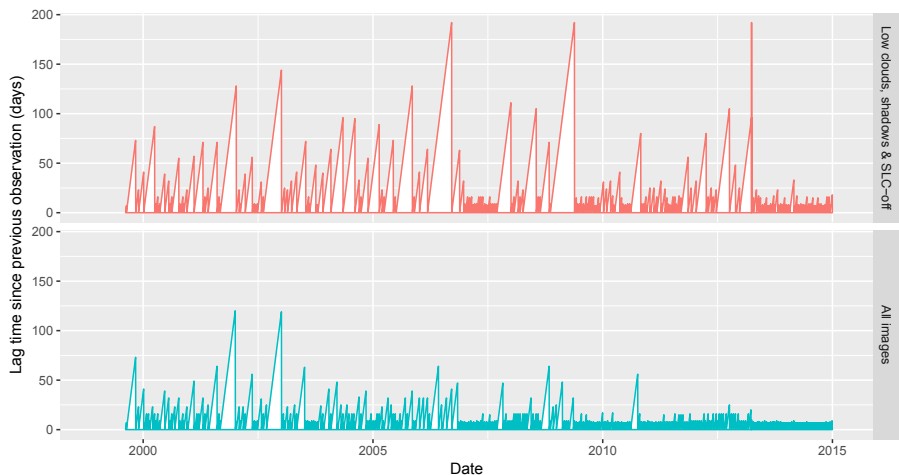

**Figure 12.** Lag time between successive Landsat images, based on all images available and after optimisation to reduce clouds, shadows and SLC-off interferences

the heterogeneous reflectance observed and the resulting classification difficulties. The influence of shadows is more moderate and less frequent, responsible for overestimations in only 5% of images. This results partly from the greater difficulties in discriminating shadows and the overlap from clouds. Pixel loss from SLC failure varies across Landsat tiles and for a given lake, 30% of the 287 Landsat 7 SLC-off images suffered from minor pixel loss (< 10%). SLC-off pixel loss led to a systematic
5  underestimation of flooded areas, on average by 35% (Fig. 11).

### 3.2.2   Optimising image availability vs. interferences

Optimising on $R^2$ tends towards removing all cloud & SLC interferences, and therefore reduces the number of available observations. Considering the objective of hydrological monitoring and maintaining sufficient temporal repetitivity, images with up to 25% SLC-off pixels and 40% cloud & shadow pixels over the studied lake cell were retained here. These thresholds

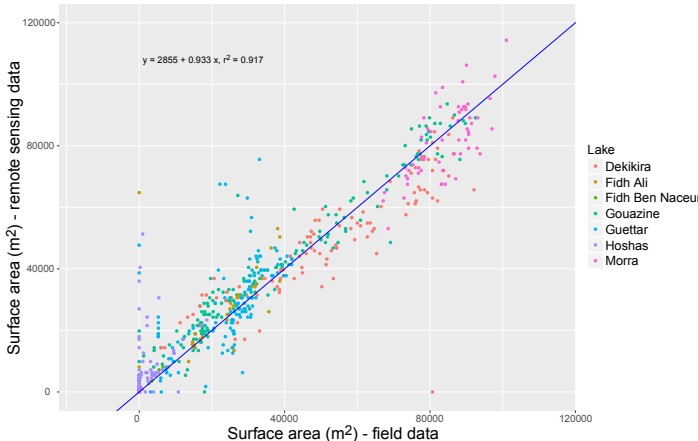

**Figure 13.** Remotely sensed surface areas with MNDWI on Landsat imagery versus field data over 1999-2014 across all 7 lakes

were found to minimise the mean squared error on surface area aggregated over 15 years which gives importance to both the number and quality of the Landsat observations over time.

This calibration removed a significant 51% of images, leaving on average 282 ±27 Landsat images suitable for observations across these 7 small reservoirs (Table 3). This equates to 1.5 images/month over the 1999-2014 period studied though these are unevenly distributed as seen in Fig. 12, due partly to low overall cloud cover between March 2013 - December 2014 over our area of interest. The acquisition cycle of Landsat 7 is also staggered by 8 days compared to Landsat 5 and 8, leading to a revisit frequency up to 8 days. The Merguellil catchment is located at the overlap of two paths visited at 7 days interval, further increasing availability.

Significantly when studying one lake, 59% of the Landsat 7 post-2003 images were discarded due to SLC-off pixels, and 31% of all Landsat images were removed due to excessive cloud presence. Variations between lakes in the number of images available were high when using the same thresholds, due to greater cloud presence on higher altitude lakes (Morra) as well as the irregular SLC interference detailed earlier. These confirm the value of lake-specific cloud detection to account for the scale of small reservoirs (i.e. a 0.1 $km^2$ area within a 33 300 $km^2$ Landsat 8 tile, i.e. 0.0003%). In other catchments, proximity to coastal areas and more pronounced orography may increase the proportion of images affected by clouds.

### 3.3 Remote monitoring of surface water dynamics in small reservoirs

Results from this semi-automated method applied across 546 Landsat images (1999-2014) and 7 lakes yielded a strong correlation ($R^2$ = 0.92) between remotely sensed flood areas and field data (Fig. 13). Confirming its excellent performance in terms of accuracy and stability across reservoirs and over several years, results also substantiate the method's ability to be used across multiple sensors notably Landsat 8 OLI, with Landsat 7 ETM+ and Landsat 5 TM, in line with what Vogelmann et al. (2001) showed on TM and ETM+. On larger lakes with long term and accurate time series such as Gouazine, $R^2$ reached 0.95 with an

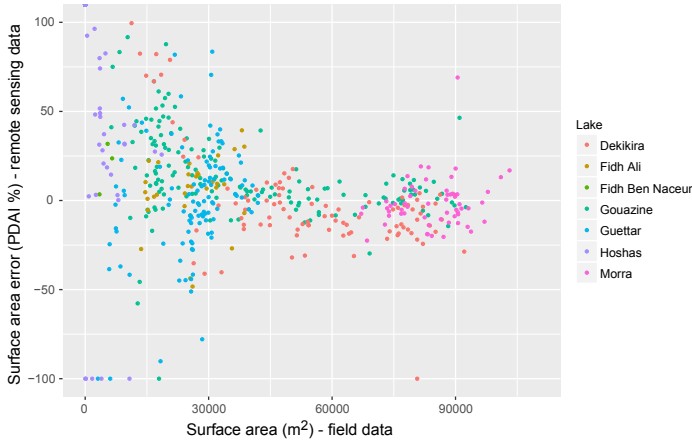

**Figure 14.** Surface area error (in PDAI %) from remote sensing data against field data for 7 lakes

RMSE of 6800 $m^2$ (Table 3). When accounting for the lake size, this equates to a median relative (PDAI) error of 11% (Fig. 14).

Landsat observations provided here accurate insights into the surface water extent dynamics, closely reproducing flood peaks and declines notably (Fig. 15). Reduced image availability due to clouds, shadows and SLC-off however prevented accurate

representation (interpolation) of the steep flood rises associated with intense rapid floods as opposed to the more gradual declines, easier to interpolate. As seen in Fig. 10 and Fig. 11, increasing the allowance of clouds, shadows and SLC-off pixels to raise image availability leads to a decline in $R^2$ and a rise in PDAI. Conversely, single outliers remain, but reducing cloud, shadows and SLC thresholds was shown not to remove these larger classification errors. These outliers are due to the inherent uncertainties in the method, partly incomplete detection of clouds and classification difficulties (e.g. overestimation as a result

of detecting water flow in the tributaries or peripheral flooding due to cells being defined broadly to account for exceptional flood volumes). Stage-derived surface areas are subject to rising uncertainty after 2008 due to the obsolescence of the rating curve (estimated at 2300 $m^2$ from Fig. 3) but Fig. 16 does not signal a marked shift in the rating curve.

### 3.3.1 Greater errors on small lakes

On smaller lakes, the influence of incorrectly detected pixels is proportionally greater due to the low spatial resolution and

reduced pixel numbers. Mean PDAI rises from 13% to 27% on surface areas under 3 ha and scattering is reflected in the reduced NSE skill values (Fig. 14 and Table 3). On Hoshas, this is exacerbated by the numerous empty periods and the greater mathematical influence on $R^2$ of minor classification noise. Errors are also proportionally greater at the lower part of the H-S rating curves. Figure 16 highlights on Hoshas the multiple classification errors (overestimation and nondetection) and the observed dispersion confirms that errors are only partially affected by the absence of prior rating curves. Over a reduced period

(2011-2014) where the lake was regularly flooded and data confidence is greater, $R^2$ for Hoshas rises to 0.77. Smaller lakes also suffer from greater classification difficulties, discussed in calibration of indices, due to more heterogeneous pixels where

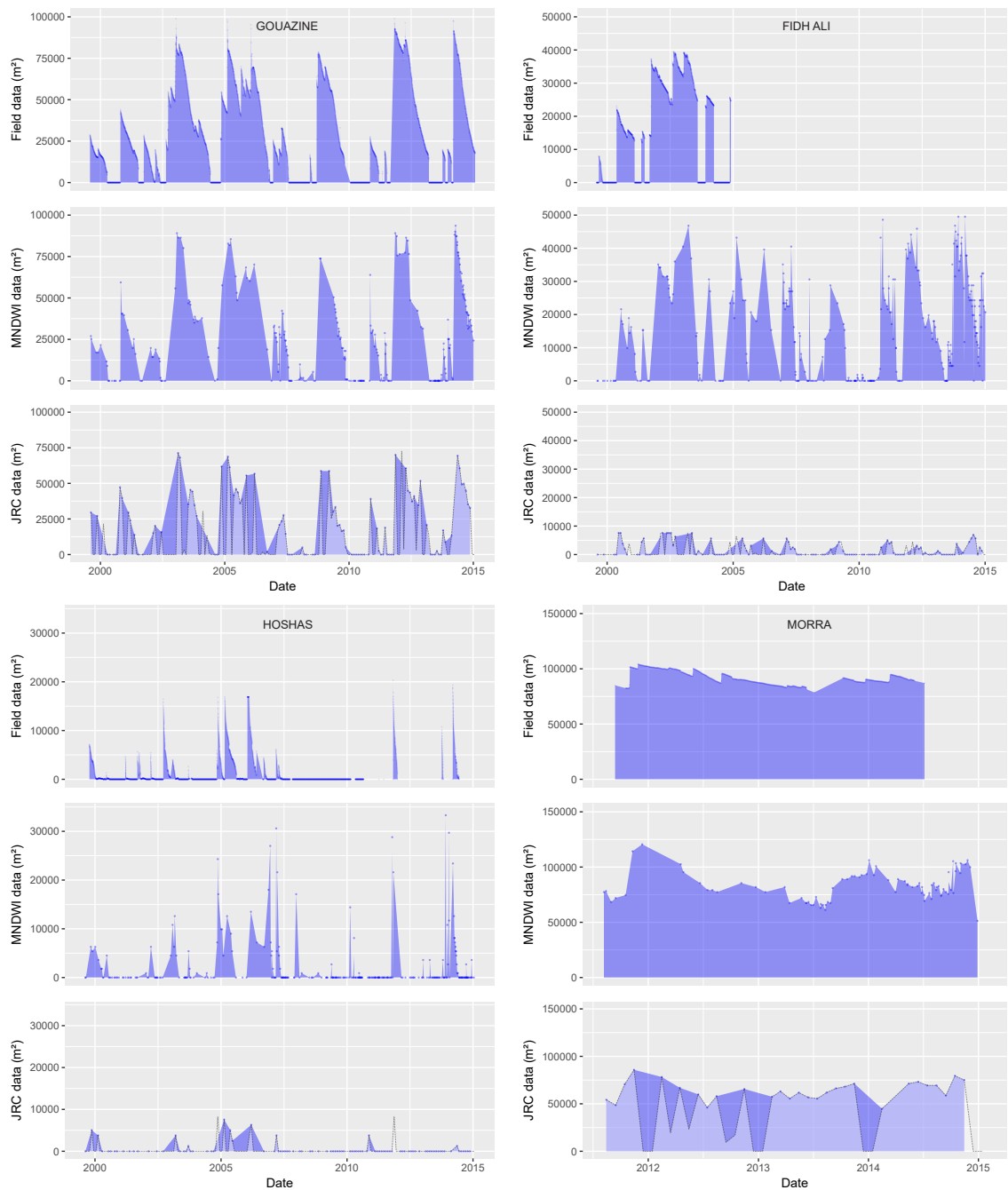

**Figure 15.** Surface water extent dynamics from field data and remote sensing over 1999-2015 for 4 lakes. JRC data (Pekel et al., 2016) is filtered to images with less than 25% NA over the lake cell (the complete JRC datasets are represented in pale blue)

soil, vegetation and water reflectances overlap. Threshold calibration of our MNDWI index led here to a minor overestimation for the lowest water levels, as seen in Fig. 14 and for Fidh Ali and low flood levels in Gouazine in Fig. 13. In absolute terms, however, mean error remains contained to 9300 $\text{m}^2$ for both small and larger lakes.

### 3.3.2 Dealing with rapid flood declines

On these smaller lakes, the method continues in many cases to be able to derive time series reproducing surface water dynamics, notably highlight drought periods and the minor flooded areas (e.g. Fidh Ali in Fig. 15). However, the reduced amplitude of the flood and crucially the rapid decline can lead to difficulties. Out of 7 events on the smallest lakes (Fidh Ben Nasseur and Hoshas), 2 minor events (0.5 ha flood) were completely undetected and two were detected after surface area had declined by 50%. On these lakes, 1 ha floods were completely infiltrated and evaporated within 21 days. Figure 15 also illustrates the

difficulty in interpolating points with variable and non negligible time lag into a hydrologically coherent daily time series. Available observations remained however relatively accurate on Fidh Ben Nasseur ($R^2$ = 0.91), showing the potential of the method even on the smallest lakes (< 1 ha), subject to sufficient temporal resolution. The launch of Sentinel-2A in 2015 and Sentinel-2B in 2017 provide increased opportunities to monitor with high temporal resolution rapid flood dynamics in small reservoirs, discussed in section 3.6.

### 3.3.3 Reduced surface water amplitudes

On the largest lake studied here (Morra), NRMSE remains moderate (14%) however the low $R^2$ translates the difficulties in extracting consistent flood trends from the remote sensing data (Fig. 15). GPS contours revealed a 9000 $\text{m}^2$ error associated with the stage-surface rating curve (Fig. 3) which has not been updated for over 20 years due to the lake remaining flooded through the year and the access difficulties to perform bathymetric surveys. Comparing remotely sensed surface areas with field

stage readings (Fig. 16) illustrates the potentiel of using Landsat observations to define and correct hypsometric relationships in small water bodies, such as Gouazine and Morra. On Hoshas, the dispersion in the values indicate that errors are only marginally influenced by the absence of prior rating curves and confirm the presence of remote sensing uncertainties. On Morra, Fig. 16 illustrates the inadequacy of the rating curve, and notably its inferior slope which leads to reducing the amplitude of flooded surface area variations. A shift in the rating curve is also apparent during the late 2000s, resulting either from a major silting

event or from a shift in the placement of the ladder. Using the Landsat observations after 2011 to define a new rating curve leads to a closer match between stage converted surface area and remotely sensed surface area. This is expected considering the bias it introduces, but here RMSE reduces only marginally by 10%, indicating that the rating curve errors are not sufficient to explain the low performance observed. This may be partly explained by the morphology of this large deep lake, where the uncertainties of the remote sensing method (combined with the rating curve) are significant compared to the amplitude of the

surface area variations (80 000 - 100 000 $\text{m}^2$ on Morra against 0 - 90 000 $\text{m}^2$ on Gouazine). Accordingly, it is not only the spatial resolution and number of pixels but the amplitude of variations in the flooded surface area over time which determine by how much remote sensing of surface water extent is affected by uncertainties and scattering.

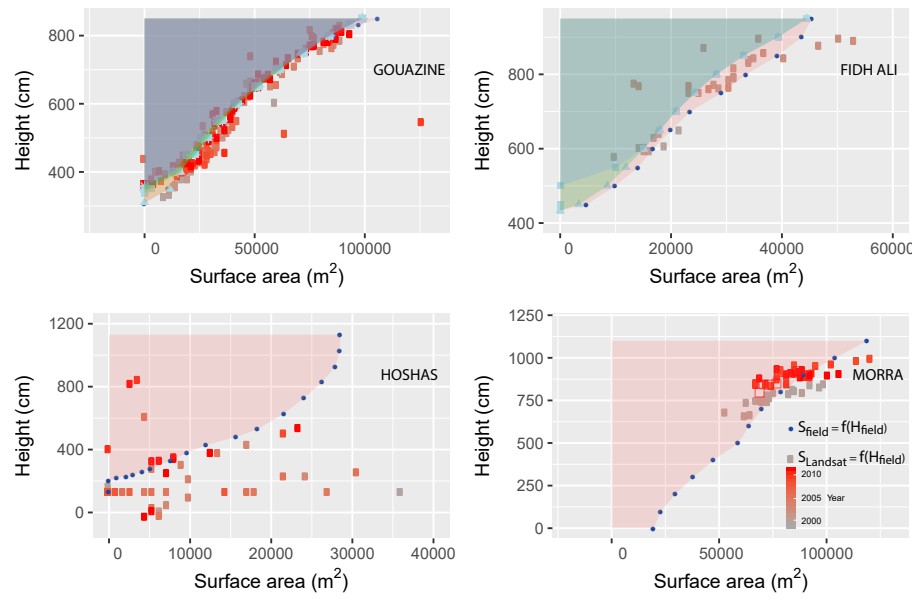

**Figure 16.** Comparison of field based H-S rating curves and the H-S rating curves that could be established based on Landsat surface water observations over 1999-2014

**Table 3.** Landsat RS performance on daily surface areas and mean annual volumes

| Lakes (observation period) | Initial capacity ($10^3 \text{m}^3$) | Mean surface area ($\text{m}^2$) | Nb images (1999-2014) | Daily surface area | | | Mean annual volume | |
|---|---|---|---|---|---|---|---|---|
| | | | | NSE | RMSE ($\text{m}^2$) | Median PDAI (%) | NSE | RMSE ($\text{m}^3$) |
| Gouazine (1999-2014) | 237 | 28700 | 269 | 0.95 | 6800 | 11.1 | 0.91 | 14400 |
| Dekikira (1999-2008) | 219 | 53900 | 240 | 0.73 | 12200 | 13.3 | 0.59 | 36600 |
| Morra (2011-2014) | 705 | 84200 | 250 | 0.43 | 12200 | 7.3 | 0.29 | 87900 |
| Guettar (2011-2014) | 150 | 24000 | 302 | 0.69 | 9600 | 18.6 | 0.81 | 8500 |
| Fidh Ali (1999-2005) | 134 | 15800 | 308 | 0.81 | 6900 | 10.4 | 0.73 | 13100 |
| Hoshas (1999-2014) | 130 | 1300 | 310 | 0.18 | 7200 | 100 | 0.38 | 5000 |
| Fidh Ben Nasseur (1999-2001) | 47 | 800 | 298 | 0.91 | 900 | 27.7 | NA | 600 |

## 3.4 Remote assessment of long term water availability in small reservoirs

Daily time series of surface area aggregated and converted into mean annual surface volumes are shown in Fig. 17. The fit with the mean volume assessed from stage data is again excellent for Gouazine ($R^2$ = 0.91) (Table 3). On other lakes, the absence of stage data over several years limited the number of values which could be compared, but for available years

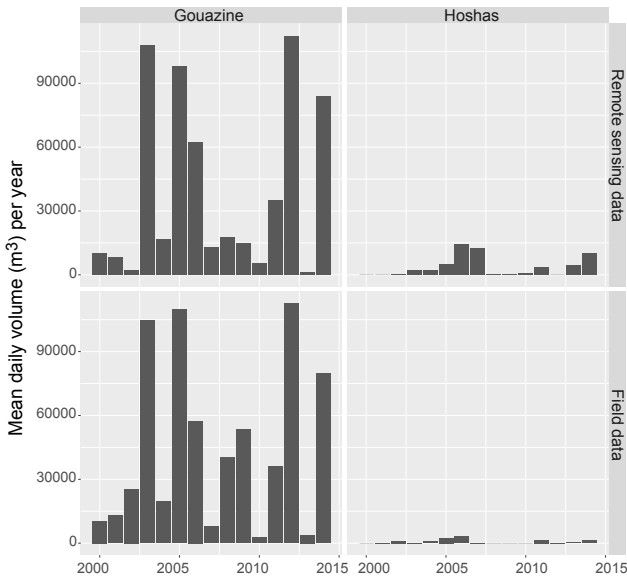

**Figure 17.** Annual mean volumes from field data and remote sensing data over 1999-2015 for 2 lakes

Fig. 18 indicates a good overall fit ($R^2$ = 0.82). The method therefore provides a valuable tool to assess and compare water resources between years and lakes. Transposed to multiple ungauged lakes where no hypsometric relationships are available, this approach can highlight the significant variability in terms of inter-lake and inter-annual water availability (Ogilvie et al., 2016a). This information may then help farmers and stakeholders quantify the risk of agricultural drought and optimise their agricultural practices and investments accordingly. In the Merguellil catchment, water availability patterns are being spatially confronted with agricultural production to explore to what extent hydrological constraints suffice in explaining agricultural water patterns and what additional socio-economic factors must be accounted for.

$R^2$ values increased for certain lakes as a result of the smoothing of the method's uncertainties (e.g. Guettar and Hoshas), but conversely reduced on other lakes (e.g. Dekikira). Interpolation over time between discrete Landsat observations can lead single outliers to reduce overall performance if they are not rapidly corrected by a subsequent observation. The influence of outliers is then variable and depends on the lag between successive correct observations. Similarly, all observations do not carry the same importance when assessing mean annual volumes. Landsat observations close to the peak of an event are accordingly more important than observations during dry periods or the (gradual) decline of flooded areas, which can be more easily interpolated.

### 3.5   Comparison with published global data sets

Analysis of available monthly surface water datasets (Pekel et al., 2016) against field data for 7 lakes over up to 15 years (1999-2014) highlighted firstly the incidence of NA pixels and the necessity to assess and remove NA pixels above individual lakes to reduce underestimations (Fig. 19) and reproduce coherent flood dynamics (Fig. 15). Filtering out images with over 25% NA values per lake cell led to optimal RMSE errors and is coherent with the SLC threshold determined with our approach. After

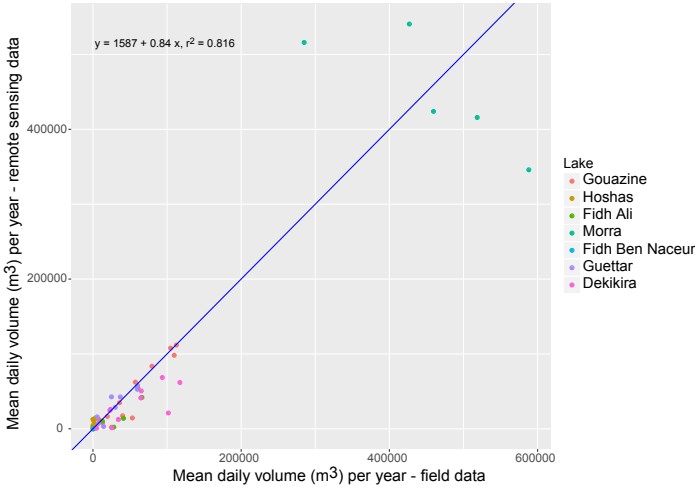

**Figure 18.** Mean daily volume per year with MNDWI on Landsat imagery versus field data over 1999-2014 across all 7 lakes

NA removal, NSE across all 7 lakes rose from 0.39 to 0.78 but Fig. 20 confirms the residual difficulties, consistent with the omission errors identified on single JRC water history rasters (Fig. 7 and Fig. 9). Mean RMSE reaches $22\,500\ \mathrm{m}^2$ and removing all NAs only improves RMSE further to $21\,800\ \mathrm{m}^2$ compared to $9300\ \mathrm{m}^2$ with the MNDWI approach. On small lakes under 5 ha which present greater shallow waters and flooded vegetation, PDAI errors are proportionally more important and reach 61%

on average. On Fidh Ali (Fig. 15), difficulties due to the significant vegetation in the lake lead to vast underestimations and fail to detect certain floods completely. On surface areas over 5 ha, PDAI remains high around 35% and interestingly continue to underestimate surface water areas. These result from omission errors during summer months due to vegetation growth in permanent lakes as well as undetected vegetated areas on the fringes inundated during large floods. Figure 15 highlights the valuable insights into the general flood dynamics that can be extracted from the JRC datasets on larger lakes such as Gouazine

and Morra, though omission errors remain apparent in the underestimated flood peaks.

These classification inaccuracies are further exacerbated by the limited number of observations. After NA removal, JRC monthly water history sets provide for instance 99 exploitable images over 1999-2014 on Gouazine, compared to 269 when using the complete L5, L7 and L8 archives filtered for clouds, shadows and SLC failure (Fig. 11 and Fig. 12). The greater temporal resolution allows better detection of dry spells, multiple flood peaks, timing of flood peaks and more coherent flood

rises and declines as seen on Gouazine, while on the small Hoshas lake two floods in 2013-2014 are otherwise undetected with JRC datasets (Fig. 15). These difficulties also have implications on the accuracy of maximum extent estimates, and though median underestimation is 41%, the spread in the error (Fig. 20) even for larger lakes, prevents systematic corrections. Surface area inaccuracies on small lakes in JRC datasets also degrade water resource assessments for farmers or water balances for hydrologists, with for instance Gouazine RMSE on mean water availability increasing by 24% to $17\,800\ \mathrm{m}^3$.

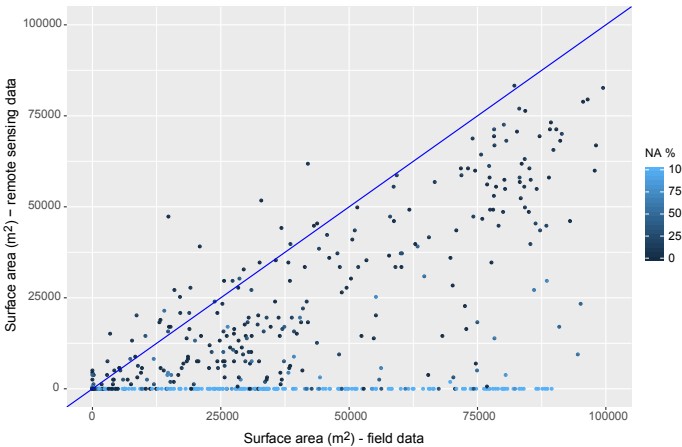

**Figure 19.** Remotely sensed surface areas with global surface water data sets (Pekel et al., 2016) versus field data over 1999-2014 across all 7 lakes. Percentage of NA pixels was calculated over each lake cell

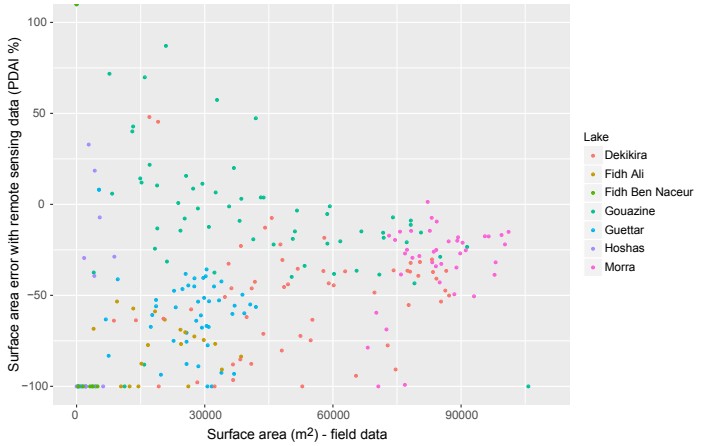

**Figure 20.** Surface area error (in PDAI %) from global surface water data sets (Pekel et al., 2016) against field data for 7 lakes

## 3.6 Comparison with Sentinel-2 imagery

Sentinel-2 imagery available on the Merguellil catchment since December 2015 was used to monitor surface water variations on Gouazine lake. Figure 21 illustrates the improved monitoring of the rise of the unique flood over 2015-2017 from combining Landsat and Sentinel-2 data. This results partly from the increased frequency of observations, but Landsat 7 and Landsat 8 were here disproportionately affected by the presence of clouds between early October and the flood in late December 2016: 3 cloudless observations out of 8 vs. 9 out of 12 for Sentinel-2. This gap in water surface observations would lead, in the absence of complementary local hydrometeorological data, to gross errors in monitoring the timing of the flood rise. The Landsat 7 observation closest to the flood peak also argues for applying multi-sensor approaches to maximise the number of surface

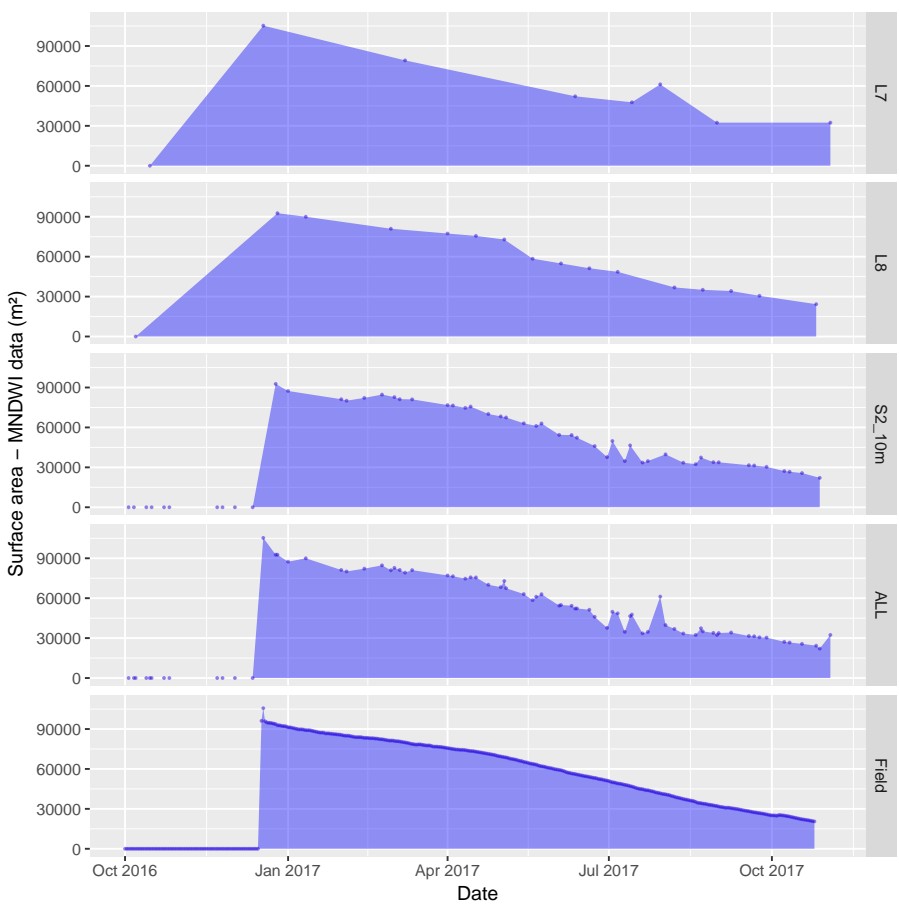

**Figure 21.** Surface water extent dynamics from Sentinel-2 and Landsat 7, 8 and field data over 2016-2017 on lake Gouazine

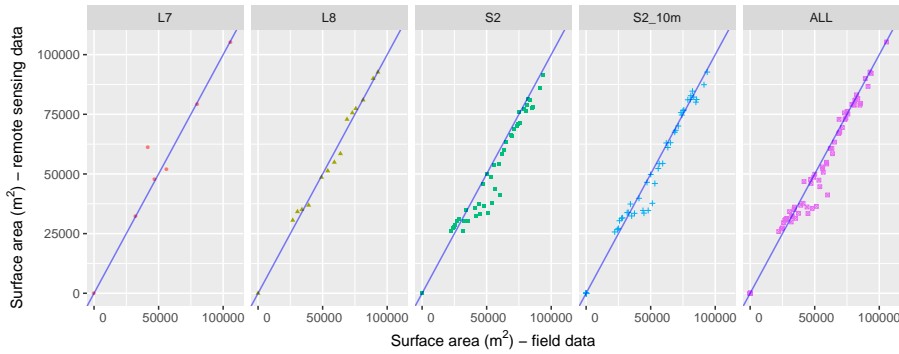

**Figure 22.** Remotely sensed surface areas with Sentinel-2 and Landsat 7, 8 data versus field data over 2016-2017 on lake Gouazine

water observations especially in small water bodies, where flood variations are sudden, and less suited to interpolation than large wetlands with gradual flood rises. The increased temporal resolution of Sentinel-2 also allows greater accuracy across the flood decline and overall RMSE across the period reaches 6900 $m^2$ compared to 23 700 $m^2$ with Landsat 7 and 8 imagery. Combining Landsat and Sentinel-2 observations further reduces RMSE to 5200 $m^2$ (NRMSE = 16.5%) and NSE rises to 0.97. Furthermore, MNDWI calculated at 20 m and at 10 m after pansharpening of the SWIR band highlights the additional improvements (Fig. 22) from the increased spatial resolution (RMSE reduces from 7800 $m^2$ to 6900 $m^2$ at 10 m), coherent with results in Du et al. (2016). The benefits from 10 m imagery are expected to be greater on smaller flooded areas, and should be further quantified based on additional in-situ monitoring of small water bodies.

## 4   Conclusions

Validated against significant hydrometric field data, this paper demonstrates the potential and limitations of Landsat imagery to monitor surface water extent dynamics and water availability within the smallest reservoirs (1-10 ha). Results confirmed the superiority of MNDWI out of six widely used band ratio indices to detect flooded areas in seven reservoirs of varying sizes and over three flood stages. Overall accuracy rates were high despite difficulties of shallow water and flooded vegetation and the MNDWI threshold remained sufficiently stable to allow automation across several lakes from 1-12 ha and over Landsat 5, 7 and Landsat 8 images. The performance of other water detection indices when used in semi automation (fixed thresholds) reduced significantly due to their greater sensitivity to minor changes in thresholds. With MNDWI, surface area errors remained low, under 15% on 5 lakes including the smallest ones, remarkable considering the demanding objective of detecting areas under 1 ha with 30 m spatial imagery.

Applied over 546 images, results highlight the impact of spatial inaccuracies as well as cloud & shadow interferences over small lakes. Residual errors from cloud uncertainties, due to undetected cirrus clouds and shadows, as well as greater vegetation, shallow water and associated mixed pixels led to a number of outliers which can be detrimental notably on smaller lakes. Filtering images with excessive cloud and SLC presence reduced temporal resolution to 1.5 images/month which can lead to minor floods and those subject to high infiltration to be overlooked by Landsat time series. With a mean RMSE of 9300 $m^2$ and $R^2$ reaching 0.9, the method however confirmed the operational potential of Landsat imagery to study surface water dynamics of floods over 3 ha, as well as long term water availability ($R^2$ = 0.82).

Comparisons with global water data sets (Pekel et al., 2016) notably confirmed the benefits and relevance of specific, complementary approaches to improve classification in smaller water bodies. Designed and calibrated to account for the specificities of small reservoirs, this semi-automated MNDWI approach reduced RMSE by 57% in comparison, by minimising omission errors from flooded vegetation and shallow waters. Accurate estimates of the total flooded surface area are notably important when converting lake surfaces to volumes (Busker et al., 2018) to quantify hydrological variables. Results also argue for the importance of exploiting all Landsat archive imagery to maximise temporal resolution, capture flood peaks, reproduce coherent flood dynamics and improve water resource assessments considering the rapid flood declines in small lakes. Clearer distinction of clouds and SLC-off in the JRC datasets would have allowed finer filtering of images due to the non systematic cloud error

observed here. Continued research in improving cirrus cloud & shadows detection and notably specific methods capable of addressing the incremental change even on small land use objects remains essential (Zhu and Woodcock, 2014).

Results also provide evidence of the rising opportunities from remote sensing in small-scale hydrology, secured by the next generation of optical sensors providing free images every 5 days (Sentinel-2) with high spatial resolution (10-20 m). Greater

spatial accuracy from 10 m pansharpened bands notably reduced RMSE by 12% on the 2016-2017 floods compared to MNDWI on 20 m Sentinel-2 bands. Combining Landsat and Sentinel-2 imagery here reduced RMSE to 5200 $m^2$ from 23 700 $m^2$ on the 2016-2017 flood thanks to frequent observations and better detection of the flood peak. Monitoring flood dynamics with optical sensors remains however dependent on low cloud cover and alternate optimisation of clouds & SLC-off or concomitant imagery sources including active sensors (e.g. Sentinel-1) could also be used to maintain more observations at critical stages

such as flood peaks (Eilander et al., 2014). Remotely sensed observations may also be combined with field data and modelling to overcome image availability issues and problems of incoherent hydrological dynamics.

Using free medium resolution Landsat imagery, this method may be replicated regionally or globally to provide long term data sets across small water bodies. Future research will help clarify whether the apparent stability of the MNDWI threshold across these 7 lakes and in other regions (Ogilvie et al., 2015) allows the approach to be transposed without additional field

calibration and whether automatic thresholding approaches may be complementary (Vala and Baxi, 2013; Coltin et al., 2016). Accurate long term water availability assessments are notably required to inform farmers of water resources in the millions of small reservoirs worldwide (Wisser et al., 2010; Lehner et al., 2011) and improve hydrological modelling. The spatialised information of the volumes captured by small lakes can be used to improve hydrological modelling in the lake catchments, especially runoff estimation due to poorly detected rainfall in semi-arid areas. Similarly, data assimilation techniques to combine

the Landsat-derived flooded volumes can improve semi-distributed hydrological modelling of larger catchments. Reducing the uncertainties on the volumes captured by small lakes is essential to improve watershed management, e.g. to assess groundwater recharge from small lakes and to clarify the cumulative influence of these water conservation works in reducing downstream runoff.

*Code and data availability.*  Landsat L1T data are available from USGS Earth Explorer. Monthly water history rasters (Pekel et al., 2016) are

available through Google Earth Engines, image collection ID: JRC/GSW1_0/MonthlyHistory. Fmask is available at https://github.com/prs021/fmask. Details on the additional scripts and R packages used to process the Landsat imagery are provided in Appendix A. Sentinel-2 surface reflectance data is available for selected regions of the world from https://theia.cnes.fr.

## Appendix A:  Remote sensing preprocessing

The following preprocessing chain was applied to 596 Landsat images over 1999-2017. Functions belonging to the R *rgdal*,

*raster*, *sp*, and *landsat* packages were used but their code was adapted to account for the differences in Landsat 8 bands and metadata files.

## A1 Radiometric normalisation

Quantized calibrated (QCAL) digital numbers (DN) for each pixel were converted to at sensor (a.k.a. top of atmosphere, TOA) spectral radiance $L$ by reversing the calibration process used by USGS (Chander et al., 2007). The linear transformation used (Eq. (A1)) can be rewritten as Eq. (A2) as $DN_{min}$ is 0 and $DN_{max}$ is 255. $G_{rescale}$ and $B_{rescale}$ are the "radiance multiplicative" and "radiance additive" rescaling factors, sometimes called gain and bias (or offset). These sensor and band specific factors were extracted automatically for each band and image from the Landsat metadata files.

$$(DN - DN_{min}) \cdot (L_{max} - L_{min}) = (DN_{max} - DN_{min}) \cdot (L - L_{min}) \tag{A1}$$

$$L = G_{rescale} \cdot DN + B_{rescale} \tag{A2}$$

where

$$G_{rescale} = \frac{L_{max} - L_{min}}{DN_{max} - DN_{min}} = \frac{L_{max} - L_{min}}{255} \tag{A3}$$

$$B_{rescale} = L_{min} - \frac{L_{max} - L_{min}}{DN_{max} - DN_{min}} DN_{min} = L_{min} \tag{A4}$$

## A2 Atmospheric corrections

At sensor radiance were then converted to at canopy (at surface) reflectance using the Dark Object Subtraction method (Chavez, 1996). The DOS approach encompasses conversion to at sensor reflectance to account for intra-annual differences in earth to sun distance and solar illumination (Paolini et al., 2006). It also incorporates atmospheric corrections to estimate top of canopy (TOC) reflectance (Eq. (A5)) and account for the complex distorting effect of air molecules, aerosol particles, water vapour, carbon dioxide, methane, etc. which absorb and scatter electromagnetic waves (Hagolle et al., 2015). Absolute corrections were preferred over relative atmospheric correction methods such as the pseudo invariant features (PIF) and histogram matching as these do not account for the band specificity of atmospheric disturbances and are therefore not recommended with band ratio classifications (Lu et al., 2002). Furthermore, key inputs for physically based models of (part of the) atmospheric effects such as aerosol optical thickness were not available from nearby Aeronet sites for historical Landsat 5 & 7 data. DOS methods provide a simple, widely used method for atmospheric corrections, proven to perform well and as reliably as other algorithms and methods (Paolini et al., 2006; Lu et al., 2002; Song et al., 2001).

$$\rho_{TOC.DOS} = \frac{(L_{sat} - L_{haze}) \cdot \pi \cdot E_{sun}{}^2}{T_{auv} \cdot (Ex_{sun} \cdot cos(90 - sun_{elev}) \cdot T_{auz} + E_{down})} \tag{A5}$$

$$L_{haze} = SHV - L_{sat1\%} \tag{A6}$$

$$L_{sat1\%} = 0.01 \cdot \frac{E_{sun} \cdot cos(90 - sun_{elev})}{\pi \cdot E_{sun}^2} \tag{A7}$$

$$L_{haze} = G_{rescale} \cdot DN_{min} + B_{rescale} - 0.01 \cdot \frac{E_{sun} \cdot cos(90 - sun_{elev})}{\pi \cdot E_{sun}^2} \tag{A8}$$

The DOS method supposes that certain objects within an image are dark (in complete shadow) and that radiance values measured for these objects are due to atmospheric scattering (path radiance, $L_{haze}$) (Chavez, 1996). These dark objects are selected from the lowest DN value across at least 1000 pixels. This defines the starting haze value (SHV) which is converted from DN to TOA radiance according to Eq. (A2). As per Eq. (A8), an allowance is made considering that pixels are rarely completely dark and 1% natural reflectance is subtracted. $L_{haze}$ converted to satellite reflectance is then subtracted to reflectance values across the image (Eq. (A5)). The improved DOS employed here calculates $L_{haze}$ for all bands using only one band (blue band) to "correlate haze values and maintain the spectral relationship between bands" (Goslee, 2011; Chavez, 1996). In the SWIR bands, where minimal scattering occurs, SHV is defined as 0 (Goslee, 2011).

$T_{auv}$ is the atmospheric transmittance from the target to the sensor, $T_{auz}$ is the atmospheric transmittance from the sun to the target and $E_{down}$ is the downwelling diffuse spectral irradiance at the surface. DOS1 level of correction which assumes no atmospheric transmission losses in either direction ($T_{auv}$ & $T_{auz}$=1) and no diffuse downward radiation at the ground ($E_{down}$=0), i.e. corrects only for additive scattering was used here. More complex corrections (for haze, aerosols) are less adapted to normalising successive images as required here and do not improve overall accuracies (Song et al., 2001; Goslee, 2011).

$E_{sun}$ is the earth to sun distance in astronomical units (AU) provided in the metadata files for OLI. For other sensors (ETM+, TM), this was calculated based on the date of acquisition indicated in the metadata (and function ESDIST in the R Landsat (Goslee, 2011) package).

$sun_{elev}$ is the solar elevation angle in degrees for the scene centre provided by the metadata file and again based on known changes in earth-sun geometry over the months. The complementary angle ($90-sun_{elev}$) provides the local solar zenith angle.

$Ex_{sun}$ is the band-specific exoatmospheric solar irradiance constant in $\mathrm{Wm}^{-2} \times \mu\mathrm{m}^{-1}$ calculated based on Eq. (A9). For OLI, $L_{max}$ and $REF_{max}$ are provided for each band. For ETM+ and TM sensors, $Ex_{sun}$ for each band were taken from the literature (Table A1).

$$Ex_{sun} = \frac{L_{max} \cdot (\pi \cdot E_{sun}^2)}{REF_{max}} \tag{A9}$$

### A3 Topographic normalisation

Finally, topographic corrections to account for the effect on surface reflectance values of reduced illumination from inclined surfaces were implemented. The Minnaert algorithm (Minnaert, 1941), shown to perform as well or better than C correction,

**Table A1.** Exoatmospheric solar constant per band for ETM+ and TM sensors (Chander et al., 2009)

| | Landsat 7 ETM+ | Landsat 5 TM |
|---|---|---|
| 1 | 1997 | |
| 2 | 1812 | 1796 |
| 3 | 1533 | |
| 4 | 1039 | |
| 5 | 230.8 | 220 |
| 6 | | |
| 7 | 84.9 | |
| 8 | 1362 | |

Cosine correction and Gamma correction (Vanonckelen et al., 2013) was used. Minnaert adds a band specific constant $K$ to the cosine correction, a trigonometric correction defined in Eq. (A10) where $\rho_t$ is the reflectance value generated by the inclined surface, and $\rho_h$ is the value on a theoretical flat surface. Illumination is modelled as per Eq. (A11) where $\gamma_i$ is the incident angle, slope angle is $\theta_p$, solar zenith angle is $\theta_z$, solar azimuth angle $\phi_a$, and aspect angle $\phi_o$. Slope and aspect were derived in

R from the SRTM version 3 30 m digital elevation model, and information on solar azimuth and zenith angle were extracted from satellite and solar geometry in the metadata files.

$$\rho_h = \rho_t \cdot \left(\frac{cos\theta_z}{IL}\right)^K \tag{A10}$$

$$IL = cos\gamma_i = cos\theta_p\, cos\theta_z + sin\theta_p\, sin\theta_z\, cos(\phi_a - \phi_o) \tag{A11}$$

**A4   Cloud and shadow detection per lake**

Fmask (Zhu and Woodcock, 2012) which uses top of atmosphere reflectances for all visible and IR bands, and brightness temperature for band 6 (Eq. (A12)) as inputs was used to assess clouds and shadow presence on each image and over each lake cell. It reports a 96.41% overall accuracy vs 84.8% for the Automated Cloud Cover Assessment (ACCA) algorithm (Irish et al., 2006) used originally by the USGS. DN values in Landsat thermal bands were converted to TOA radiance using Eq. (A2) and then to (brightness) temperature (Kelvin) using Eq. (A12).

$$brightness\ temperature = \frac{K_2}{log(\frac{K_1 \cdot \varepsilon}{L} + 1)} \tag{A12}$$

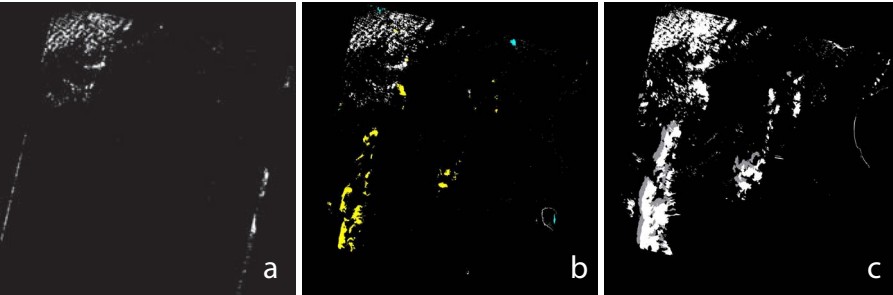

**Figure A1.** Comparing cloud detection methods on Landsat 8 29.03.2013 image: (a) Clouds detected over 1.7% of the image with ACCA(Irish et al., 2006) and Goslee (2011) methods; (b) Clouds incl. cirrus detected over 8.2% of the image with Landsat 8 1380nm band; (c) Clouds incl. cirrus and cloud shadows detected over 11.5% of the image with Fmask (Zhu and Woodcock, 2014) method.

$K1$ and $K2$ can be found in the Landsat 8 metadata and in the literature for Landsat 5 and 7 (Chander et al., 2009). $\varepsilon$ is emissivity, and can be estimated as 0.95 to account for atmospheric interferences. The method then combines probability masks with a series of rule based tests to identify potential cloud pixels based on the cloud spectral behaviour. Cloud shadows are modelled based upon estimating cloud height based on temperature, view angle and solar illumination geometry reported
in the metadata files (Huang et al., 2010; Zhu and Woodcock, 2012). In addition to cloud shadows, Fmask version (3.2.1) used here also exploits the Landsat 8 1380 nm wavelength for the detection of the cirrus clouds (Fig. A1). This wavelength present on other lower spatial resolution sensors is strongly absorbed by the water vapour in the (lower) atmosphere therefore only high cirrus clouds where water vapour is low will reflect solar radiation in this wavelength. As discussed, associated methods combining the information from several (15 for Tmask) successive images to improve cloud shadow detection (Zhu
and Woodcock, 2014; Goodwin et al., 2013) were not suitable, due to the localised and rapid changes investigated here.

*Competing interests.*   The authors declare that they have no conflict of interest.

*Acknowledgements.*   We gratefully acknowledge the collaboration from the Direction Générale de l'Aménagement et de la Conservation des Terres Agricoles (DG ACTA) and Direction Générale des Ressources en Eau (DGRE) in Tunis and local representatives at the Kairouan and Siliana Commissariat Régional au Développement Agricole (CRDA). These works were partly financed by the ANR AMETHYST and
SICMED DYSHYME projects. Sentinel-2 surface reflectance data was obtained as part of the CNES THEIA Sentinel-2 project for Tunisia. SPOT 5 10 m imagery was provided through CNES ISIS programme, request 703.

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
