# Peer review of "Surface water monitoring in small water bodies: potential and limits of multi-sensor Landsat time series"

_Hydrology and Earth System Sciences, 2018_

## Referee Comment (RC1) · Anonymous Referee #1 · 22 Feb 2018

General comments

This paper presents an evaluation of the LANDSAT capability to monitor small water bodies in a Central Tunisia. An extensive and accurate evaluation is carried based on a precious dataset extending over fifteen years which makes this work valuable. However, more work should be done for this study to bring an original contribution over previous published literature, since no new methodological developments are implemented neither novel findings form a hydrological point of view are reported.

Several issues should be addressed before publication in HESS. The main points are highlighted below:

[Figure]

1- The last few years have seen new developments and products release in optical remote sensing that are not addressed by the current study, although they could be quite relevant to the final objective of monitoring small water bodies with a rapid temporal dynamic in time.

First, Sentinel 2 data are available since the end of 2015 with a revisit time of 10 days and 5 days since the launch of Sentinel 2B last year. Recent works have shown the capability of Sentinel 2 (alone or in combination to Landsat) to monitor small water bodies using spectral indexes over different regions (i.e. Kaplan et al. 2017, Du et al. 2016 and Zohu et al. 2017). Addressing the potential of Landsat alone, as done in the current study, does not allow to take into account the potential of the multi-sensor combination available with the actual generation of optical satellite sensors.

Second, land surface reflectances and cloud mask for the Landsat sensors are available since few years now (Landsat level-2 data). It would be quite interesting to take into account this widely used product in the current study and analyse its impact on the water bodies classification employed in this work.

2- It is important to discuss the value of this work for hydrological applications: beyond the assessment of the capability of monitoring water areas in the study region (that could be further investigated, see point 2 about it) is the analysis of water areas derived by Landsat over 15 years leading to any new finding on the hydrology of this region (in addition to those reported by Olgivie et al 2016b)?

3- The comparison to the JRC product is a bit misleading since the Peckel database concern "open water" only, while the classification carried out for this paper also includes flooded vegetated area. For the comparison to be meaningful, open water pixels only should be considered. It is however quite interesting to evaluate the proportion of vegetated/flooded area, not taken into account by the JRC database, over open water areas for these small water bodies. To do this, the authors could attempt at classifying separately open water and water with vegetation: would this be possible? (I

guess calibration/validation could be more difficult if this information is not reported in the in-situ data base)

4- Finally, several points need to better explained or clarified concerning both the methodology and the in-situ measurements (see specific comments below). In particular, a point that needs clarification is the methodology used to derive water areas in-situ: for what I understand for calibration and validation water contours were derived by GPS, but for the long term analysis water levels coupled to bathymetry data were used. If this the case, more details on the hypsometric relationships applied should be given and an accuracy assessment of water areas derived in this way should be carried out. When the bottom is quite flat (which can happen for flooded areas during the rainy season), small changes in water level can result in significant changes in water area.

Specific comments:

Abstract: line 6: better small instead of smallest

Pg 2 line 31: 16 days since the 1970s? Prior to Landsat7/8 data are generally much less frequent

Pg 3 lines 8-12: this is not very clear given that several studies (including the cited Peckel et al. 2016, Olgivie et al 2016b and Jones et al 2017) analysed the long term dynamics of water bodies including small water bodies

Pg 3 line 20: the term "low resolution" is a bit confusing. Does it refer to Landsat 30m resolution? Or to medium resolution sensors like the cited MODIS?

Pg 3 last par: temporality issues can be now better addressed by combining to Sentinel2 (i.e. Kaplan et al. 2017, Du et al. 2016 and Zohu et al. 2017), see general comment above

Pg 5 section 2.2. this section should be more clearly written (see point 4 above): how many GPS contours were available? And more important: how in situ areas for the long term analysis were derived? (line 13 refers to water volumes derived from stage

values, what about areas?)? An accuracy assessment on the in-situ measurements would be more than welcomed!

Section 2.5: employing different metrics allows a complete evaluation of Landsat performances. However the manuscript is a bit confusing about it (minimum RMSE is used to define the thresholds, PDAI for validation purposes, RMSE and NSE for the long term analysis). A table summarising all the metrics employed would help the reader. For completeness, RMSE could be also reported in Table 2 and mean PDAI in table 3.

Fig 3: this example shows one of the biggest lake analysed. Given the paper focus on small water bodies, it would be interesting to add some examples of MNDWI performances for smaller lakes, and discuss this in term of the amount of vegetated and/or mixed pixels.

Fig. 12 This figure would be more informative if the authors could add the information on the MNDWI points directly derived from Landsat data and those interpolated

Section 3.3.3 lines 21-24: As already pointed out this should be better clarified in the methodology section. An error analysis seems necessary.

Section 3.5 see point 3 in general comments. Fig. 15 Given that the JRC dataset only concern "open water" a 1:1 line should not be expected

Conclusion pg 23, line 32 reference to SWOT is not appropriate given the focus on small water bodies. SWOT mission spec are indeed given for water bodies with area above 1 km2 Line 33: Low cloud: not if radar is used (i.e. Sentinel1)

Appendix A: see point 2 in the general comments concerning the Landsat land surface products

Technical comments:

Fig 7: white dots are difficult to see, please change the color table

[Figure]

[Figure]

---

## Referee Comment (RC2) · C. Cudennec (Referee) · 26 Feb 2018

Ogilvie et al. develop and present an important methodology which values available remote sensing informations and opens low cost applications in the future. It allows assessing and mapping filling of reservoirs spread across a semiarid area, with strong implications for monitoring resources availability across the territory where water is crucial in the Water-Food-Energy-Development nexus; as well as agregation of hydrological impacts on the functioning of the whole basin. The methodology is developed and tested thanks to accurate field campaigns, over a pilot basin in central semiarid Tunisia which is emerging as a strong reference since 15 years.

[Figure]

The methodology is based on the use of 7 well known spectral indices; and their assessment thanks to the available Landsat images is well justified and discussed, against field difficulties such as shallow waters, vegetation development, frequency of images regarding rapid dynamic of floods.

Litterature review about these 7 indices, and their previous applications and limits could be expanded in section 2.4 as it is too implicit so far. It is refered to a "widely used" status whereas it is mentioned at some places that these are more or less relevant (rural gently sloping watershed, design to detect mixed water reflectance, MNDWI outperformed AWEI when extracting narrow water bodies...).

Also a rapid positioning of this approach regarding spatial altimetry and geodesy applied to lakes would be welcome to better assess the challenges of small reservoirs in data scarce regions (See for instance Cretaux et al., 2011, 2015, 2016).

My major concern is the need to better explicit the hydrometric approach: remote sensing allows to assess water surface. Deducing water storage / volume / availability / resources needs the use of a volume-stage-surface / rating / bathymetric curve. This should be made more explicit in principle in the Introduction and then the Methods section. Further, the bathymetric curves of small reservoirs in this region are not stationnary, as erosion-silting is important, yet very heterogeneous across regions and so reservoirs. The Authors say page 19 that the curve of the biggest reservoir (Mora) is obsolescent because not updated over the past 20 years. This points the need to precisely address the issue of the exact availability and accuracy of bathymetric curves for every considered reservoirs (beyond the short statement on P5, L15 referring to old Albergel and Rejeb, 1997 reference); as well as consequences in terms of uncertainties in the overall method.

Minor issues: - P1, L22: Reservoirs do not reduce soil loss - but sediment transfer once in the network. - P2, L20: Okavango and Mekong Deltas. - P3, L33: Localise instead of localised. - Section 2.5: What are the exact dates of the images - and what

are the characteristics of the rainfalls over that particular period? - P9, L17: Provide references about Gouazine basin and reservoir (Nasri et al., for instance). - P13, L6: Duplication of "in". - P16, L2: The Merguellil catchment. - P33, L4: author Calvez duplicated.

Christophe Cudennec

---

## Author Comment (AC1) · 13 Apr 2018

**Reply to C. Cudennec - Referee #2**

We wish to thank C. Cudennec for the valuable and relevant comments provided. Please find below a point by point reply to the issues raised and the steps taken to modify the manuscript. References not in the additional manuscript have been specified here. We look forward to finalising the revised manuscript based on these comments and additions.

**Ogilvie et al. develop and present an important methodology which values available remote sensing informations and opens low cost applications in the future. It allows assessing and mapping filling of reservoirs spread across a semiarid area, with strong implications for monitoring resources availability across the territory where water is crucial in the Water-Food-Energy-Development nexus; as well as agregation of hydrological impacts on the functioning of the whole basin. The methodology is developed and tested thanks to accurate field campaigns, over a pilot basin in central semiarid Tunisia which is emerging as a strong reference since 15 years. The methodology is based on the use of 7 well known spectral indices; and their assessment thanks to the available Landsat images is well justified and discussed, against field difficulties such as shallow waters, vegetation development, frequency of images regarding rapid dynamic of floods.**

**Litterature review about these 7 indices, and their previous applications and limits could be expanded in section 2.4 as it is too implicit so far. It is refered to a "widely used" status whereas it is mentioned at some places that these are more or less relevant (rural gently sloping watershed, design to detect mixed water reflectance, MNDWI outperformed AWEI when extracting narrow water bodies...).**

This section has been modified to provide additional detail of the applications and specificities of the spectral water indices. The following text has been inserted in the manuscript:

"Six widely used water detection indices (Table 1) were compared to assess their suitability to monitor floods on small reservoirs in terms of detection accuracy and threshold stability. These include NDWI (Normalised Difference Water Index) developed by McFeeters (1996) exploiting the difference in reflectance of water in the green and NIR bands. Xu (2006) proposed a modified NDWI (MNDWI) where the NIR band was substituted by the SWIR band to improve distinction of built up features over water. In parallel, Lacaux et al. (2007) developed the NDPI (Normalised Difference Pond Index) which also exploits the low reflectance of water in SWIR and its contrast with the green band. NDPI is effectively the opposite of MNDWI and was therefore not investigated separately. Gao (1996) developed a NDWI but using NIR and SWIR bands. Xu (2006) later defined this as NDMI (Normalised Difference Moisture Index). NDTI (Normalised Difference Turbidity Index) was also developed using the red and green bands, and exploits the principle that turbid water reacts like bare soils, i.e. low reflectance in green and high in red. The NDVI (Normalised Difference Vegetation Index) (Rouse et al. 1973) is one of the most well known band ratio index which exploits the contrast between the peak reflectance in the infrared band and the low reflectance in the red to monitor vegetation. It has also been used to detect water bodies (Ma et al. 2007; Mohamed et al. 2004) as the index becomes positive in presence of vegetation and negative for water bodies. Finally, AWEI (Automated Water Extraction Index) (Feyisa et al. 2014) was empirically developed to discriminate water over several large lakes using wavelengths within 5 Landsat bands. Two variants exist, AWEInsh and AWEIsh, the latter being optimised to remove

(urban and topographic) shadow pixels. Both can be used in succession in the presence of highly reflective areas (snow, roofs, etc.) but in this rural, semi-arid gently sloping watershed, the AWEInsh was used. For SWIR, band 5 for TM/ETM+ and band 7 for OLI were used in accordance with other studies (Ji et al. 2009; Ouma and Tateishi 2006). For NVDI and NDTI, values below the threshold were classified as water, and conversely for the four other indices."

**Also a rapid positioning of this approach regarding spatial altimetry and geodesy applied to lakes would be welcome to better assess the challenges of small reservoirs in data scarce regions (See for instance Cretaux et al., 2011, 2015, 2016).**

This relevant comment has led us to add the following text in the introduction of the manuscript.

"Satellite altimetry originally developed to monitor the ocean's surface has increasingly been exploited and adapted since the 1990s to monitor height variations of continental surface waters (Cretaux et al., 2016). Used across rivers and large lakes, recent works have sought to transpose these to smaller water bodies (from 50 ha) but highlight several limitations relating essentially to the poor density of altimetry tracks, the long along-track path lengths, and low revisit times (Baup et al., 2016, Avisse et al. 2017). Altimeters aboard Topex, Envisat, Jason-1 and Jason-2 have temporal resolutions ranging from 10 to 35 days and high vertical accuracy however their narrow swaths and long along track path lengths (1 km) restrict their application essentially to large lakes (> 100 km2, Avisse et al., 2017). Other altimeters have improved spatial resolution and cover more of the globe but at the expense of low revisit times (368 days for Cryostat-2), removing any monitoring possibilities. The Dahiti altimetry database (Schwatke et al., 2015) employed by Busker et al., 2018 combines the tracks of numerous altimeters to optimise the sites covered, temporal resolution and the length of observations. These provide data for 168 sites in Africa, however none in Tunisia and lakes must have a minimum 300 m diameter (circa 7 ha). The Sentinel-3a and 3b constellation provide major improvements in their along track resolution (300 m) making them potentially suitable for lakes around 4 ha but inter-tracks of 52 km mean many lakes are not covered by their trajectories (Cretaux et al., 2016). Furthermore, radar altimetry provides an estimate of the altitude of the water surface, based on the two-way travel time of the radar pulse and the known altitude of the satellite, but assessing absolute volumes requires site specific data such as stage height or topographic data (Baup et al., 2016). Avisse et al., 2017 showed that DEM data taken before lakes were built or when it was empty could be used but on larger lakes (59 ha to 379 ha) with 30 m data."

Crétaux, J. F., Abarca-del-Río, R., Berge-Nguyen, M., Arsen, A., Drolon, V., Clos, G., & Maisongrande, P. (2016). Lake volume monitoring from space. Surveys in Geophysics, 37(2), 269-305.

Schwatke, C., Dettmering, D., Bosch, W., & Seitz, F. (2015). DAHITI–an innovative approach for estimating water level time series over inland waters using multi-mission satellite altimetry. Hydrology and Earth System Sciences, 19(10), 4345-4364.

Busker, T., de Roo, A., Gelati, E., Schwatke, C., Adamovic, M., Bisselink, B., Pekel, J.-F., and Cottam, A.: A global lake and reservoir volume analysis using a surface water dataset and satellite altimetry, Hydrol. Earth Syst. Sci. Discuss., in review, 2018.

**My major concern is the need to better explicit the hydrometric approach: remote sensing allows to assess water surface. Deducing water storage / volume / availability / resources needs the use of a volume-stage-surface / rating / bathymetric curve. This should be made more explicit in principle in the Introduction and then the Methods section. Further, the bathymetric curves of small reservoirs**

**in this region are not stationnary, as erosion-silting is important, yet very heterogeneous across regions and so reservoirs. The Authors say page 19 that the curve of the biggest reservoir (Mora) is obsolescent because not updated over the past 20 years. This points the need to precisely address the issue of the exact availability and accuracy of bathymetric curves for every considered reservoirs (beyond the short statement on P5, L15 referring to old Albergel and Rejeb, 1997 reference); as well as consequences in terms of uncertainties in the overall method.**

Both reviewers rightly highlighted that further details on the bathymetric curves, their evolution over time from silting and the associated uncertainties are required here. To assess the long-term performance of Landsat imagery to quantify flooded surface areas and volumes, stage data converted using site specific hypsometric (stage-surface-volume) relationships were used. These were acquired over 1990-2007 through previous research projects and additional levelling of Hoshas as part of this research in 2014. A figure has been added to the manuscript to explicit the number of relationships for each lake. To overcome the absence of regular surveys on some lakes (e.g. Morra), silting was modelled based on research on silting in 15 lakes in and around the Merguellil catchment (Albergel et al., 2003). These showed that the decline in capacity over time from silting could be modelled through linear regression. Analysis of these 70 rating curves highlighted the progressive shift in the parameters of the rating curve power relation ($V = B * S^{beta}$). Beta is shown to increase gradually over time, reflecting the decreasingly concave nature of the lakes floor. The evolution over time of the site-specific power relations was therefore calculated based on a gradual annual increase of the beta parameter and an associated decrease in maximum capacity ($V_{max}$). Initial Vmax were here known based on the inventories and used to calculate the initial $S_{max}$. By supposing that $S_{max}$ at the spillway does not evolve over time, which is acceptable based on the rating curves in our possession, the resulting B is then calculated over time. In practice, silting is heterogeneous and occurs through sudden, discrete events not a linear, incremental process but local studies confirmed the difficulties in modelling sediment transport in these small catchments (Hentati et al., 2010).

These additional details on the hypsometric relationships applied will be added to the manuscript. Furthermore, accuracy assessments of power relations updated over time to account for silting against the available updated hypsometric rating curves (as per Ogilvie et al. 2016b) as well as additional GPS contours acquired on Morra and Guettar in 2014 will be integrated. The potential to use regular Landsat derived surface area estimates at multiple water levels to create and correct the site-specific hypsometric relationships will also be discussed.

Hentati, A., Kawamura, A., Amaguchi, H., & Iseri, Y. (2010). Evaluation of sedimentation vulnerability at small hillside reservoirs in the semi-arid region of Tunisia using the Self-Organizing Map. Geomorphology, 122(1-2), 56-64.

**Minor issues:**

**-P1, L22: Reservoirs do not reduce soil loss -but sediment transfer once in the network.**

This sentence has been modified as : "These have been built to reduce sediment transfer and silting of downstream dams, as well as harvest scarce and unreliable water resources for local users (Habi and Morsli, 2011 ; Wisser et al., 2010)"

**-P2, L20: Okavango and Mekong Deltas.**

The text has been modified accordingly.

**-P3, L33: Localise instead of localised.**

The text has been modified accordingly.

**-Section 2.5: What are the exact dates of the images -and what are the characteristics of the rainfalls over that particular period?**

The precise dates of the images (29.03.2013, 24.05.2013, 09.06.2013) have been included in table 2. The following sentence has also been added: "The gradual decline in water surface area observed on all lakes across these 3 images are also coherent with the rainfall characteristics. Rainfall recorded over this period (March-June 2013) ranged between 27 mm and 44 mm across the lakes concentrated on 1 event on the 24.04.2013."

**-P9, L17: Provide references about Gouazine basin and reservoir (Nasri et al., for instance).**

Two references have been added here. "Gouazine lake (Nasri et al., 2004, Al Ali et al., 2008) which possessed both the longest time series (over 15 years) and the most accurate data and rating curves (updated 6 times) was used to optimise the thresholds."

**-P13, L6: Duplication of "in".**

The text has been modified accordingly.

**-P16, L2: The Merguellil catchment.**

The text has been modified accordingly.

**-P33, L4: author Calvez duplicated.**

The text has been modified accordingly.

---

## Author Comment (AC2) · 13 Apr 2018

**Reply to anonymous Referee #1**

We wish to thank the anonymous reviewer for the valuable, thoughtful comments provided. Please find below a point by point reply to the issues raised and the steps taken to modify the manuscript. References not in the additional manuscript have been specified here. We look forward to finalising the revised manuscript based on these comments and additions.

**General comments**

**This paper presents an evaluation of the LANDSAT capability to monitor small water bodies in a Central Tunisia. An extensive and accurate evaluation is carried based on a precious dataset extending over fifteen years which makes this work valuable. However, more work should be done for this study to bring an original contribution over previous published literature, since no new methodological developments are implemented neither novel findings form a hydrological point of view are reported.**

We fully agree that part of the strength of this manuscript lies in the long term hydrometric field data for small reservoirs. This allows extensive evaluation of remote sensing methods and specifically their capacity to reproduce hydrological process, here the variability of flood dynamics and long-term water availability in these small water bodies (some of which represent only a few Landsat pixels). As reflected in reviewer #2 comments, the paper does also seek to compare, adapt and optimise available methodologies to the specificities of small reservoirs, notably in terms of lowering water detection index thresholds to capture the shallow waters and waters with standing vegetation, common in small lakes. Furthermore, it develops a suitable approach to maximise image availability whilst minimising the influence of clouds, shadows and SLC off interferences to maintain sufficient temporal resolution and accuracy and capture the rapid flood dynamics in small lakes. We appreciate the relevant suggestions by the reviewer and complementary work is being integrated into the manuscript as detailed below.

**Several issues should be addressed before publication in HESS. The main points are highlighted below:**

**1-The last few years have seen new developments and products release in optical remote sensing that are not addressed by the current study, although they could be quite relevant to the final objective of monitoring small water bodies with a rapid temporal dynamic in time.**

**First, Sentinel 2 data are available since the end of 2015 with a revisit time of 10 days and 5 days since the launch of Sentinel 2B last year. Recent works have shown the capability of Sentinel 2 (alone or in combination to Landsat) to monitor small water bodies using spectral indexes over different regions (i.e. Kaplan et al. 2017, Du et al. 2016 and Zohu et al. 2017). Addressing the potential of Landsat alone, as done in the current study, does not allow to take into account the potential of the multi-sensor combination available with the actual generation of optical satellite sensors.**

The reviewer rightly highlights the raised potential for hydrological monitoring offered by new products such as those offered by the combination of Sentinel-2a and Sentinel-2b. Our ongoing research indeed seeks to quantify the benefits in terms of spatial accuracy, flood dynamics and water availability of Sentinel-2 over Landsat across water bodies of different sizes and flood dynamics in the

Sahel. In this paper, however, we had chosen to focus on the potential of Landsat as for diachronic studies pre-2015 or for long term monitoring as here, Landsat imagery remains the most adequate, freely available source of imagery with albeit limited, spatial and temporal resolution. Nevertheless, this does not detract from the interest and relevance of presenting within this paper results obtained when combining Landsat and Sentinel-2 data.

For this purpose, additional field data for the main lake (Gouazine, where monitoring has continued) has been obtained up to the end of 2017. Additional Landsat 7 and 8 imagery has been acquired and processed using the same treatment chain (radiometric and topographic corrections to surface reflectance, MNDWI water detection, Fmask cloud and shadow detection, etc.). Sentinel 2 data over our region of interest is available since December 2015 and surface reflectance products from the THEIA Sentinel-2 project for the Merguellil catchment over 2015-2017 have been acquired. This additional data is being processed and will be integrated in the manuscript to provide an evaluation of the improvements in terms of NSE and daily surface RMSE and mean annual volume RMSE on small lakes from the multi-sensor combination (L7 ETM+, L8 OLI, S2 MSI) and increased spatial resolution (20m and 10m with pansharpening of SWIR on S2)

**Second, land surface reflectances and cloud mask for the Landsat sensors are available since few years now (Landsat level-2 data). It would be quite interesting to take into account this widely used product in the current study and analyse its impact on the water bodies classification employed in this work.**

Landsat surface reflectance products (L2A) from ESPA indeed became available during our research. As a result, these were not used at the time and the relevant corrections were programmed as per the steps defined in the appendix A. To better inform future users, Landsat 8 L2A ESPA on-demand imagery for the Merguellil catchment for 2013-2015 has now been acquired and flooded areas extracted using MNDWI to be compared with the 13 DGPS contours (to remove uncertainties/errors from hypsometric relations). Results show a marked difference as RMSE reaches 21 200 m2 with the ESPA surface reflectance products compared to 3 200 m2 with surface reflectance products through our treatment chain. Results reveal greater difficulties on the smaller lakes but lowering the threshold further to increase the flooded vegetation leads to overestimations on the large lakes. These additional results are being integrated into the revised manuscript and discussed in light of the different atmospheric corrections used (6S-LaSRC, cf. Vermote et al., 2016; Doxani et al., 2018) and the importance of finer topographic corrections (GCM DEM 1km vs. 30m in our method).

The cloud detection algorithm used by USGS is now based on Cmask (Foga et al., 2017), a C version of the same Fmask method we implemented. As a result, the results from using the cloud and shadow values from the level 1 Quality Assessment (QA) bands are identical to those using the Fmask algorithm. The description of cloud and show detection in the Methods is being updated to reflect this.

Vermote, E., Justice, C., Claverie, M., & Franch, B. (2016). Preliminary analysis of the performance of the Landsat 8/OLI land surface reflectance product. Remote Sensing of Environment, 185, 46-56.

Doxani, G., Vermote, E., Roger, J. C., Gascon, F., Adriaensen, S., Frantz, D., ... & Louis, J. (2018). Atmospheric Correction Inter-Comparison Exercise. Remote Sensing, 10(2), 352.

Foga, S., Scaramuzza, P.L., Guo, S., Zhu, Z., Dilley, R.D., Beckmann, T., Schmidt, G.L., Dwyer, J.L., Hughes, M.J., Laue, B. (2017). Cloud detection algorithm comparison and validation for operational Landsat data products. Remote Sensing of Environment, 194, 379-390.

**2-It is important to discuss the value of this work for hydrological applications: beyond the assessment of the capability of monitoring water areas in the study region (that could be further investigated, see point 2 about it) is the analysis of water areas derived by Landsat over 15 years leading to any new finding on the hydrology of this region (in addition to those reported by Olgivie et al 2016b)?**

This research has indeed led to additional insights into the hydrology of this region. Ogilvie et al 2016b illustrate the high interannual and interlake variability identified after developing and transposing the approach across 51 ungauged lakes. The water availability patterns were then confronted with agricultural surveys, questionnaires and interviews to explore to what extent hydrological constraints suffice in explaining agricultural water patterns and what additional socio-economic factors must be accounted for. This work is being submitted for publication. In parallel, the spatialised information of the volumes captured by small lakes is being used to improve hydrological modelling of runoff in these catchments. An Ensemble Kalman filter to integrate the Landsat derived flooded volumes with 7 hydrological models (GR4J + water balance) and improve runoff estimation due to poorly detected rainfall has been developed and this work has been submitted for publication. Finally, further research seeks to combine the Landsat-derived flooded volumes across all 51 lakes, to improve semi-distributed hydrological modelling of the Merguellil catchment. Reducing the uncertainties on the volumes captured by small lakes is notably required to improve watershed management, e.g. to assess groundwater recharge from small lakes and to clarify the cumulative influence of these water conservation works in reducing downstream runoff. These hydrological applications will be mentioned in the discussion and conclusions of the revised manuscript to better highlight the value of this approach for hydrological investigations.

**3-The comparison to the JRC product is a bit misleading since the Peckel database concern "open water" only, while the classification carried out for this paper also includes flooded vegetated area. For the comparison to be meaningful, open water pixels only should be considered. It is however quite interesting to evaluate the proportion of vegetated/flooded area, not taken into account by the JRC database, over open water areas for these small water bodies. To do this, the authors could attempt at classifying separately open water and water with vegetation: would this be possible? (I guess calibration/validation could be more difficult if this information is not reported in the in-situ data base)**

The JRC database focusses on global surface water open water but as stated in Pekel et al., 2016, this is a "Known issue and planned improvement". Their approach did not include "bodies of water […] obscured by floating, overhanging and standing vegetation", and likewise Mueller et al., 2016 highlighted the difficulties due to the "presence of water and vegetation within the pixel" and that their "product may not be fit for […] small farm dams". The interest of our comparison with JRC datasets is then to show to what extent our approach can lead to more pertinent evaluation of the flooded areas in small lakes. The text will be clarified accordingly to emphasize that the value of this approach over JRC datasets is precisely because it is capable of detecting with sufficient accuracy, both pixels with only pure water and mixed pixels with floating/standing vegetation. It is notably important to detect the total surface area when using surface-volume relations to assess the runoff volumes captured, or water availability patterns.

As the reviewer rightly states, the field data consists of total lake surface area but does not distinguish pure open water and water with floating/standing vegetation, present both on the edges and centre of water bodies. We propose however to exploit supervised classification on 10m multi-spectral SPOT

imagery available for March 2013 and May 2013 (from an CNES ISIS project) as ground truth to quantify on these 2 images to what extent the difference with the JRC datasets are due to standing/floating vegetation.

**4-Finally, several points need to better explained or clarified concerning both the methodology and the in-situ measurements (see specific comments below). In particular, a point that needs clarification is the methodology used to derive water areas in-situ: for what I understand for calibration and validation water contours were derived by GPS, but for the long term analysis water levels coupled to bathymetry data were used. If this the case, more details on the hypsometric relationships applied should be given and an accuracy assessment of water areas derived in this way should be carried out. When the bottom is quite flat (which can happen for flooded areas during the rainy season), small changes in water level can result in significant changes in water area.**

As stated by the reviewer, the calibration and validation of the spectral water indices was performed against DGPS contours to remove uncertainties/errors due to hypsometric relationships. To assess the long-term performance of Landsat imagery to quantify flooded surface areas and volumes, stage data converted using site specific hypsometric (stage-surface-volume) relationships were used. These were acquired over 1990-2007 through previous research projects and additional levelling of Hoshas as part of this research in 2014. A figure has been added to the manuscript to explicit the number of relationships for each lake. To overcome the absence of regular surveys on some lakes (e.g. Morra), silting was modelled based on research on silting in 15 lakes in and around the Merguellil catchment (Albergel et al. 2003). These showed that the decline in capacity over time from silting could be modelled through linear regression. Analysis of these 70 surface-volume rating curves highlighted the progressive shift in the parameters of the rating curve power relation ($V = B * S^{beta}$). Beta is shown to increase gradually over time, reflecting the decreasingly concave nature of the lakes floor. The evolution over time of the site-specific power relations was therefore calculated based on a gradual annual increase of the beta parameter and an associated decrease in maximum capacity ($V_{max}$). Initial Vmax were here known based on the inventories and used to calculate the initial $S_{max}$. By supposing that $S_{max}$ at the spillway does not evolve over time, which is acceptable based on the rating curves in our possession, the resulting B is then calculated over time. In practice, silting is heterogeneous and occurs through sudden, discrete events not a linear, incremental process but local studies confirmed the difficulties in modelling sediment transport in these small catchments (Hentati et al., 2010).

These additional details on the hypsometric relationships applied will be added to the manuscript. Furthermore, accuracy assessments of power relations updated over time to account for silting against the available updated hypsometric rating curves (as per Ogilvie et al. 2016b) as well as additional GPS contours acquired on Morra and Guettar in 2014 will be integrated. The potential to use regular Landsat derived surface area estimates at multiple water levels to create and correct the site-specific hypsometric relationships will also be discussed.

Hentati, A., Kawamura, A., Amaguchi, H., & Iseri, Y. (2010). Evaluation of sedimentation vulnerability at small hillside reservoirs in the semi-arid region of Tunisia using the Self-Organizing Map. Geomorphology, 122(1-2), 56-64.

**Specific comments:**

**Abstract: line 6: better small instead of smallest**

The text has been rectified accordingly.

**Pg 2 line 31: 16 days since the 1970s? Prior to Landsat7/8 data are generally much less frequent**

The text has been rectified as follows to better clarify this issue: "Landsat which provides free multispectral images since the 1970s at medium geometric resolution (30 m since Landsat 4 in 1982) every 16 days (since Landsat 7 in 1999, as previous sensors present multiple gaps) therefore continues to provide the most potential to detect and monitor small water bodies."

**Pg 3 lines 8-12: this is not very clear given that several studies (including the cited Peckel et al. 2016, Olgivie et al 2016b and Jones et al 2017) analysed the long term dynamics of water bodies including small water bodies**

This section has been reorganised to better clarify the novelty of the respective research papers. Jones et al., 2017 as Liebe et al., 2005 focussed on small lakes to map reservoirs and/or provide a snapshot of floods at certain dates (seasonality or maximum extent), but did not explore their dynamics over time, which introduce further research questions relating to threshold stability, automation, and sufficient image availability to reproduce flood dynamics. As Pekel et al., 2016 state "measuring long term changes remains a challenge". Pekel et al.'s work provides a remarkable investigation of water dynamics, however their study did not focus specifically on small water bodies. Yamazaki and Trigg 2016, Mueller et al., 2016 and Yamazaki et al., 2015, who also focussed on global Landsat water inventories, recognise difficulties "due to the presence of both water and vegetation within the pixels" and that their "product may not be fit for [...] small farm dams". Pekel et al. mention omission errors of 23% on seasonal water bodies but "sample pixels of seasonal water bodies within 1° tiles" were used. These therefore do not specifically include or focus on small lakes. Considering the scarcity of ground truth data on small lakes, this is therefore a rare opportunity to specifically quantify the performance on small lakes of long term Landsat monitoring. Ogilvie et al. 2016b focussed on developing an approach for ungauged lakes, where no hypsometric relationships were available and illustrating applications on 51 lakes.

**Pg 3 line 20: the term "low resolution" is a bit confusing. Does it refer to Landsat 30m resolution? Or to medium resolution sensors like the cited MODIS?**

This paragraph refers to the work undertaken to provide long term monitoring at regional and global scale with Landsat imagery. The sentence has therefore been modified to: "These errors on small lakes are influenced by the insufficient spatial resolution but essentially by the increased presence of flooded vegetation and shallow waters which affect the reflectance signal…"

**Pg 3 last par: temporality issues can be now better addressed by combining to Sentinel2 (i.e. Kaplan et al. 2017, Du et al. 2016 and Zohu et al. 2017), see general comment above**

As stated, under the general comment 1 in this document, this paper chose to focus on Landsat's ability, considering that "Recent sensors such as Sentinel-2 capture 10 m images of the entire globe every 5 days providing enhanced opportunities but hydrological investigations which require historical perspectives will continue to rely on previous sensors", here Landsat. This section on page 3 therefore refers to the issues relating to "Landsat imagery to map and monitor small water bodies " However as stated, under general comment 1, a specific comparison of the combined benefit of using Landsat and Sentinel-2 since 2015 is being integrated to the manuscript.

**Pg 5 section 2.2. this section should be more clearly written (see point 4 above): how many GPS contours were available? And more important: how in situ areas for the long term analysis were derived? (line 13 refers to water volumes derived from stage values, what about areas?)? An accuracy assessment on the in-situ measurements would be more than welcomed!**

This section has been modified as follows : "19 GPS contours were acquired, providing a range of flooded surface areas from 0.2 ha to 7.8 ha (Table 2) to test the performance of indices."

"Stage values were converted to surface areas and water volumes using rating curves available for each lake." As described in general comment 4, further details and an accuracy assessment of the rating curves is being added to the manuscript.

**Section 2.5: employing different metrics allows a complete evaluation of Landsat performances. However the manuscript is a bit confusing about it (minimum RMSE is used to define the thresholds, PDAI for validation purposes, RMSE and NSE for the long term analysis). A table summarising all the metrics employed would help the reader. For completeness, RMSE could be also reported in Table 2 and mean PDAI in table 3.**

We indeed chose to use a variety of metrics to allow a complete and relevant evaluation of Landsat's performance. Confusion matrices were used as is common practice to quantify the classification accuracy of the water detection indices. PDAI was added to provide a directly relevant assessment of the resulting surface area error on each lake. To calibrate the clouds, shadow and SLC off thresholds (% of interference tolerated) to maximise image availability whilst reducing the detrimental influence of these interferences, RMSE over the 15 years of data were calculated and minimised (using 5% increments of each interference). The skill of the method in terms of flood dynamics and water availability were then calculated in terms of RMSE and NSE to highlight how well the results of the method fit the observed values (NSE) and in terms of the amplitude of the errors (RMSE). RMSE was not included in table 2 as these are errors on single values (i.e. each image and each lake). In table 3, we feel RMSE is more appropriate than mean PDAI, as mean PDAI would be significantly influenced by individual outliers and does therefore not provide as good an indication of mean error as RMSE. Based on the reviewer's recommendation we have gathered in table 2 all metrics relating to the calibration/validation of the water detection indices (i.e. results over 7 lakes and 3 images). Table 3 presents all the metrics relating to the 7 (different) lakes used in long term monitoring (i.e. results over 7 different lakes and 546 images).

**Fig 3: this example shows one of the biggest lake analysed. Given the paper focus on small water bodies, it would be interesting to add some examples of MNDWI performances for smaller lakes, and discuss this in term of the amount of vegetated and/or mixed pixels.**

Additional figures of the performance of the MNDWI approach against the DGPS contours on other small lakes have been added. Further to our reply under general comment 3, we propose to use the classified SPOT 10m imagery for May 2013 to distinguish open water and vegetated pixels and illustrate here which pixels are correctly classified with MNDWI and with the JRC database.

**Fig. 12 This figure would be more informative if the authors could add the information on the MNDWI points directly derived from Landsat data and those interpolated**

The figure has been modified to add points for the values of the individual Landsat observations.

**Section 3.3.3 lines 21-24: As already pointed out this should be better clarified in the methodology section. An error analysis seems necessary.**

Please see reply under general comments 4 and details on the error analysis to be included.

**Section 3.5 see point 3 in general comments. Fig. 15 Given that the JRC dataset only concern "open water" a 1:1 line should not be expected**

Please also see reply to general comment 3. This figure seeks specifically to show how JRC datasets lead to errors on small water bodies, due to their specificities (which include shallow waters, and standing/floating vegetation) which the method developed by Pekel et al. is currently unable to include. Additional comparisons have now been added to clarify these differences.

**Conclusion pg 23, line 32 reference to SWOT is not appropriate given the focus on small water bodies. SWOT mission spec are indeed given for water bodies with area above 1 km2 Line 33: Low cloud: not if radar is used (i.e. Sentinel1)**

The reference to SWOT has been removed. Line 33 has been modified to clarify that these comments refer only to optical sensors and Sentinel-1 has been specifically mentioned in line 3 page 24 after active sensors.

"Monitoring flood dynamics with optical sensors remains however dependent on low cloud cover and results here point to the value of assessing their presence at the lake and not image level."

"Alternate optimisation of clouds & SLC-off or concomitant imagery sources including active sensors (e.g. Sentinel-1) could also be used to maintain more observations at critical stages such as flood peaks (Eilander et al., 2014)."

**Appendix A: see point 2 in the general comments concerning the Landsat land surface products**

Please see reply under point 1 in the general comments.

**Technical comments:**

**Fig 7: white dots are difficult to see, please change the color table**

The figure has been modified accordingly.

---

## Author Response (AR1)

**Reply to anonymous Referee #1 – post revisions to the manuscript**

We wish to thank again the anonymous reviewer for their valuable, thoughtful comments which have helped strengthen the manuscript. As recommended, additional work to quantify the improvements from combining Landsat 7,8 with Sentinel-2 data, to compare the results with publicly distributed Landsat surface reflectance products, to assess more precisely the performance of MNDWI vs JRC data sets in open water and mixed pixels and to quantify the accuracy of the hypsometric relationships has been integrated. Please find below an updated point by point reply to the issues raised and details of the additions and modifications to the manuscript.

**General comments**

**This paper presents an evaluation of the LANDSAT capability to monitor small water bodies in a Central Tunisia. An extensive and accurate evaluation is carried based on a precious dataset extending over fifteen years which makes this work valuable. However, more work should be done for this study to bring an original contribution over previous published literature, since no new methodological developments are implemented neither novel findings form a hydrological point of view are reported.**

We fully agree that part of the strength of this manuscript lies in the long term hydrometric field data for small reservoirs. This allows extensive evaluation of remote sensing methods and specifically their capacity to reproduce hydrological processes, here the variability of flood dynamics and long-term water availability in these small water bodies (some of which represent only a few Landsat pixels). As reflected in reviewer #2 comments, the paper does also seek to compare, adapt and optimise available methodologies to the specificities of small reservoirs, notably in terms of lowering water detection index thresholds to capture the shallow waters and waters with standing vegetation, common in small lakes. Furthermore, it develops a suitable approach to maximise image availability whilst minimising the influence of clouds, shadows and SLC off interferences to maintain sufficient temporal resolution and accuracy and capture the rapid flood dynamics in small lakes. We appreciate the relevant suggestions by the reviewer and complementary work has been integrated into the manuscript as detailed below.

**Several issues should be addressed before publication in HESS. The main points are highlighted below:**

**1-The last few years have seen new developments and products release in optical remote sensing that are not addressed by the current study, although they could be quite relevant to the final objective of monitoring small water bodies with a rapid temporal dynamic in time.**

**First, Sentinel 2 data are available since the end of 2015 with a revisit time of 10 days and 5 days since the launch of Sentinel 2B last year. Recent works have shown the capability of Sentinel 2 (alone or in combination to Landsat) to monitor small water bodies using spectral indexes over different regions (i.e. Kaplan et al. 2017, Du et al. 2016 and Zohu et al. 2017). Addressing the potential of Landsat alone, as done in the current study, does not allow to take into account the potential of the multi-sensor combination available with the actual generation of optical satellite sensors.**

The reviewer rightly highlights the raised potential for hydrological monitoring offered by new products such as those offered by the combination of Sentinel-2a and Sentinel-2b. Our ongoing research indeed seeks to quantify the benefits in terms of spatial accuracy, flood dynamics and water availability of Sentinel-2 over Landsat across water bodies of different sizes and flood dynamics in the Sahel. In this paper, however, we had chosen to focus on the potential of Landsat as for diachronic studies pre-2015 or for long term monitoring as here, Landsat imagery remains the most adequate, freely available source of imagery with albeit limited, spatial and temporal resolution. Nevertheless, this does not detract from the interest and relevance of presenting within this paper results obtained when combining Landsat and Sentinel-2 data.

For this purpose, additional field data for the main lake (Gouazine, where monitoring has continued) was obtained up to the end of 2017. 50 additional Landsat 7 and 8 images were acquired and processed using the same treatment chain (radiometric and topographic corrections to surface reflectance, MNDWI water detection, Fmask cloud and shadow detection, etc.). Sentinel 2 data over our region of interest is available since December 2015 and 84 surface reflectance products from the THEIA Sentinel-2 project for the Merguellil catchment over 2015-2017 were acquired. This additional data was integrated in the manuscript to provide an evaluation of the improvements in terms of NSE and daily surface RMSE and mean annual volume RMSE on small lakes from the multi-sensor combination (L7 ETM+, L8 OLI, S2 MSI) and increased spatial resolution (20m and 10m with pansharpening of SWIR on S2).

The following text has been added in new sections under methods (section 2.9):

*"Sentinel-2A launched in June 2015 by the European Space Agency provides freely available multispectral imagery at 10 m spatial resolution (20 m in SWIR band) every 10 days. Its constellation with Sentinel-2B launched in March 2017 increases temporal resolution to 5 days. 84 surface reflectance products over 2015-2017 distributed by THEIA were used to quantify the performance of Sentinel-2 imagery and its combination with Landsat imagery after 2015 in terms of RMSE and NSE for daily surface water monitoring. 50 additional Landsat 7 and Landsat 8 images were processed to surface reflectance through our treatment chain to extract flooded areas and additional field data from ongoing in situ monitoring on 1 lake (Gouazine) was acquired.*

*THEIA Sentinel-2 products include atmospheric corrections based on the MAJA (MACSS-ATCOR Joint Algorithm) treatment chain developed by CNES-CESBIO (Hagolle et al., 2015) and German DLR. These notably include 30 m topographic corrections, as did the Landsat imagery. Intensity-Hue-Saturation (IHS) pansharpening (Carper et al., 1990) was used on SWIR bands to allow water detection at 10 m (Du et al., 2016; Kaplan et al., 2018) with MNDWI and compare performance with 20 m and 30 m MNDWI. The optimal MNDWI threshold on Sentinel-2 imagery was independently calibrated against k-means (unsupervised) classification (Jaine 2010) of flooded areas, and was substantially lower (-0.2) than with Landsat imagery (-0.09). Cloud masks provided by the MACCS algorithm were used directly, to reduce difficulties with Fmask and Sen2cor methods resulting from the absence of the thermal band."*

And under results (section 3.6) of the manuscript:

*"Sentinel-2 imagery available on the Merguellil catchment since December 2015 was used to monitor surface water variations on Gouazine lake. Figure 21 illustrates the improved monitoring of the rise of the unique flood over 2015-2017 from combining Landsat and Sentinel-2 data. This results partly from the increased frequency of observations, but Landsat 7 and Landsat 8 were here disproportionately affected by the presence of clouds between early October and the flood in late December 2016: 3 cloudless observations out of 8 vs. 9 out of 12 for Sentinel-2. This gap in water surface observations would lead, in the absence of complementary local hydrometeorological data, to gross errors in*

*monitoring the timing of the flood rise. The Landsat 7 observation closest to the flood peak also argues for applying multi-sensor approaches to maximise the number of surface water observations especially in small water bodies, where flood variations are sudden, and less suited to interpolation than large wetlands with gradual flood rises.*

*The increased temporal resolution of Sentinel-2 also allows greater accuracy across the flood decline and overall RMSE across the period reaches 6900 m² compared to 23 700 m² with Landsat 7 and 8 imagery. Combining Landsat and Sentinel-2 observations further reduces RMSE to 5200 m² (NRMSE = 16.5%) and NSE rises to 0.97. Furthermore, MNDWI calculated at 20 m and at 10 m after pansharpening of the SWIR band highlights the additional improvements (Fig. 22) from the increased spatial resolution (RMSE reduces from 7800 m² to 6900 m² at 10 m), coherent with results in Du et al., 2016. The benefits from 10 m imagery are expected to be greater on smaller flooded areas, and should be further quantified based on additional in-situ monitoring of small water bodies."*

These results were also summarised in the conclusions and abstract.

**Second, land surface reflectances and cloud mask for the Landsat sensors are available since few years now (Landsat level-2 data). It would be quite interesting to take into account this widely used product in the current study and analyse its impact on the water bodies classification employed in this work.**

Landsat surface reflectance products (L2A) from ESPA indeed became available during our research. As a result, these were not used at the time and the relevant corrections were programmed as per the steps defined in the appendix A. To better inform future users, Landsat 8 L2A ESPA on-demand imagery for the Merguellil catchment for 2013 were acquired and flooded areas extracted using MNDWI to be compared with the 13 DGPS contours (to remove uncertainties/errors from hypsometric relations).

The following text was notably added:

*"Level 2A products corrected to land surface reflectance using the Landsat Ecosystem Disturbance Adaptive Processing System (LEDAPS) for sensors TM and ETM+ or Landsat Surface Reflectance Code (LaSRC) for the OLI sensor were not available for our catchment and time period at the time of research. Two Landsat 8 LaSRC products now available from ESPA were acquired for 2013 to analyse the possibility of applying the water detection approach developed in this paper directly to level 2A imagery. Remotely sensed flooded areas were quantified and compared with 13 GPS contours to assess relative errors."*

*"Comparison between the Landsat surface reflectance products (level 2A) provided by ESPA and 13 DGPS contours revealed that lake surface areas are significantly underestimated on all lakes, except Mdinia (Fig. 8) when using the MNDWI threshold determined here (-0.09). RMSE reaches 22 000 m² with the L2A products compared to 3200 m² with surface reflectance products through our treatment chain, largely resulting from underestimation in small lakes. Lowering the MNDWI threshold further (to -0.25) to increase the classification of water with standing vegetation can reduce RMSE to 7000 m² but leads to overestimations (over 20%) on large lakes such as Mdinia. Considering the good performance of the atmospheric corrections provided by ESPA's 6S Landsat Surface Reflectance Code (6S-LaSRC) (Doxani et al., 2018; Vermote et al., 2016), the greater dispersion in the reflectance values observed may be explained by the finer topographic corrections performed in our treatment chain (30 m SRTM vs. 1 km GCM DEM by LaSRC), necessary when monitoring small water bodies."*

The cloud detection algorithm used by USGS is now based on Cmask (Foga et al., 2017), a C version of the same Fmask method we implemented. As a result, the results from using the cloud and shadow values from the level 1 Quality Assessment (QA) bands are identical to those using the Fmask algorithm. The description of cloud and show detection in the Methods has been updated as follows to reflect this:

*"Level 1 Quality Assessment bands provided by USGS now employ Cmask, a C version of this cloud detection algorithm, therefore providing comparable data (Foga et al., 2017)"*

Vermote, E., Justice, C., Claverie, M., & Franch, B. (2016). Preliminary analysis of the performance of the Landsat 8/OLI land surface reflectance product. Remote Sensing of Environment, 185, 46-56.

Doxani, G., Vermote, E., Roger, J. C., Gascon, F., Adriaensen, S., Frantz, D., ... & Louis, J. (2018). Atmospheric Correction Inter-Comparison Exercise. Remote Sensing, 10(2), 352.

Foga, S., Scaramuzza, P.L., Guo, S., Zhu, Z., Dilley, R.D., Beckmann, T., Schmidt, G.L., Dwyer, J.L., Hughes, M.J., Laue, B. (2017). Cloud detection algorithm comparison and validation for operational Landsat data products. Remote Sensing of Environment, 194, 379-390.

**2-It is important to discuss the value of this work for hydrological applications: beyond the assessment of the capability of monitoring water areas in the study region (that could be further investigated, see point 2 about it) is the analysis of water areas derived by Landsat over 15 years leading to any new finding on the hydrology of this region (in addition to those reported by Olgivie et al 2016b)?**

This research has indeed led to additional insights into the hydrology of this region. Ogilvie et al 2016b illustrate the high interannual and interlake variability identified after developing and transposing the approach across 51 ungauged lakes. The water availability patterns were then confronted with agricultural surveys, questionnaires and interviews to explore to what extent hydrological constraints suffice in explaining agricultural water patterns and what additional socio-economic factors must be accounted for. This work is being submitted for publication. In parallel, the spatialised information of the volumes captured by small lakes is being used to improve hydrological modelling of runoff in these catchments. An Ensemble Kalman filter to integrate the Landsat derived flooded volumes with 7 hydrological models (GR4J + water balance) and improve runoff estimation due to poorly detected rainfall has been developed and this work has been submitted for publication. Finally, further research seeks to combine the Landsat-derived flooded volumes across all 51 lakes, to improve semi-distributed hydrological modelling of the Merguellil catchment. Reducing the uncertainties on the volumes captured by small lakes is notably required to improve watershed management, e.g. to assess groundwater recharge from small lakes and to clarify the cumulative influence of these water conservation works in reducing downstream runoff.

These hydrological applications have now been mentioned in the discussion of results and in conclusions of the revised manuscript to better highlight the value of this approach for hydrological investigations.

*"Transposed to multiple ungauged lakes where no hypsometric relationships are available, this approach can highlight the significant variability in terms of inter-lake and inter-annual water availability (Ogilvie et al., 2016a). This information may then help farmers and stakeholders quantify the risk of agricultural drought and optimise their agricultural practices and investments accordingly. In the Merguellil catchment, water availability patterns are being spatially confronted with agricultural*

*production to explore to what extent hydrological constraints suffice in explaining agricultural water patterns and what additional socio-economic factors must be accounted for."*

*"Accurate long term water availability assessments are notably required to inform farmers of water resources in the millions of small reservoirs worldwide (Wisser et al., 2010; Lehner et al., 2011) and improve hydrological modelling. The spatialised information of the volumes captured by small lakes can be used to improve hydrological modelling in the lake catchments, especially runoff estimation due to poorly detected rainfall in semi-arid areas. Similarly, data assimilation techniques to combine the Landsat-derived flooded volumes can improve semi-distributed hydrological modelling of larger catchments. Reducing the uncertainties on the volumes captured by small lakes is essential to improve watershed management, e.g. to assess groundwater recharge from small lakes and to clarify the cumulative influence of these water conservation works in reducing downstream runoff."*

**3-The comparison to the JRC product is a bit misleading since the Peckel database concern "open water" only, while the classification carried out for this paper also includes flooded vegetated area. For the comparison to be meaningful, open water pixels only should be considered. It is however quite interesting to evaluate the proportion of vegetated/flooded area, not taken into account by the JRC database, over open water areas for these small water bodies. To do this, the authors could attempt at classifying separately open water and water with vegetation: would this be possible? (I guess calibration/validation could be more difficult if this information is not reported in the in-situ data base)**

The JRC database focusses on global surface water open water but as stated in Pekel et al., 2016, this is a "Known issue and planned improvement". Their approach does in fact seek to map pixels into one of "three target classes, either water, land or non-valid observations (snow, ice, cloud or sensor-related issues)." In practice, however many "bodies of water […] obscured by floating, overhanging and standing vegetation" are excluded, and likewise Mueller et al., 2016 highlighted the difficulties due to the "presence of water and vegetation within the pixel" and that their "product may not be fit for [...] small farm dams". The interest of our comparison with JRC datasets is to show to what extent our approach can lead to more pertinent evaluation of the flooded areas in small lakes. The text has been clarified to emphasize that the value of this approach over JRC datasets is precisely because it is capable of detecting with sufficient accuracy, both pixels with only pure water and mixed pixels with floating/standing vegetation.

As the reviewer rightly states, the field data consists of total lake surface area but does not distinguish pure open water and water with floating/standing vegetation, present both on the edges and centre of water bodies. Instead, we have exploited 10 m multi-spectral SPOT imagery available for May 2013 (from an CNES ISIS project) where NDVI maps are used to complement the ground truth data and illustrate to what extent the difference with the JRC datasets are due to standing/floating vegetation.

The following text was notably added:

*"Additionally, high spatial resolution (10 m) SPOT 5 multispectral imagery from 26.05.2013 was used to identify standing and floating vegetation on the 7 lakes based on superior NDVI (Normalised Difference Vegetation Index) values, and discuss the performance of the expert systems approach in pure and vegetated pixels. "*

*"Accurate estimates of the total flooded surface area are notably important when converting lake surfaces to volumes (Busker et al., 2018) to quantify hydrological variables."*

*"NDVI maps created from high resolution (10 m) SPOT imagery confirmed the accurate detection of pure water pixels by both methods and illustrate the difficulties with the JRC datasets where lakes include standing or floating vegetation (Fig. 9). The lake fringes are notably excluded by the JRC approach, which on larger lakes such as Morra and Mdinia leads to minor underestimation of the flooded area. However, on lake Garia S, where vegetation is observed on much of the west of the lake, this leads to substantial underestimations (PDAI = 80%). NDVI maps also highlight additional pure water pixels in the south of Garia S and En Mel lakes which are excluded by the expert systems approach. Conversely, on lake Kraroub, the JRC datasets included vegetated pixels in May 2013 but not in June 2013. Though results point largely to difficulties in detecting shallow waters where reflectance is affected by vegetation and turbidity, other factors influencing the JRC approach such as terrain shadows and lake morphology could not be excluded here. The good performance on Kraroub in May 2013 confirms that in some situations, the expert systems approach is capable of reaching its goal of detecting water even with variable ``chlorophyll concentration, total suspended solids and coloured dissolved organic matter load'' (Pekel et al., 2016), but not systematically."*

**4-Finally, several points need to better explained or clarified concerning both the methodology and the in-situ measurements (see specific comments below). In particular, a point that needs clarification is the methodology used to derive water areas in-situ: for what I understand for calibration and validation water contours were derived by GPS, but for the long term analysis water levels coupled to bathymetry data were used. If this the case, more details on the hypsometric relationships applied should be given and an accuracy assessment of water areas derived in this way should be carried out. When the bottom is quite flat (which can happen for flooded areas during the rainy season), small changes in water level can result in significant changes in water area.**

As stated by the reviewer, the calibration and validation of the spectral water indices was performed against DGPS contours to remove uncertainties/errors due to hypsometric relationships. To assess the long-term performance of Landsat imagery to quantify flooded surface areas and volumes, stage data converted using site specific hypsometric (stage-surface-volume) relationships were used. These were acquired over 1990-2007 through previous research projects and additional levelling of Hoshas as part of this research in 2014. A figure has been added to the manuscript to explicit the number of relationships for each lake, and illustrate the evolution of the rating curves as a result of localised silting over time. Furthermore, accuracy assessments of the hypsometric rating curves and their updates over time have been integrated. These notably exploit 7 additional GPS contours acquired in 2013 and 2014 on Guettar and Morra. Finally, the potential to use Landsat derived surface area estimates at multiple water levels to create and correct the site-specific hypsometric relationships is also illustrated and discussed.

The following text has been inserted in the Methods:

[revised manuscript text omitted]

**Specific comments:**

**Abstract: line 6: better small instead of smallest**

The text has been rectified accordingly.

**Pg 2 line 31: 16 days since the 1970s? Prior to Landsat7/8 data are generally much less frequent**

The text has been rectified as follows to better clarify this issue: *"Landsat which provides free multispectral images since the 1970s at medium geometric resolution (30 m since Landsat 4 in 1982) every 16 days (since Landsat 7 in 1999, as previous sensors present multiple gaps) therefore continues to provide the most potential to detect and monitor small water bodies."*

**Pg 3 lines 8-12: this is not very clear given that several studies (including the cited Peckel et al. 2016, Olgivie et al 2016b and Jones et al 2017) analysed the long term dynamics of water bodies including small water bodies**

This section has been reorganised to better clarify the novelty of the respective research papers. Jones et al., 2017 as Liebe et al., 2005 focussed on small lakes to map reservoirs and/or provide a snapshot of floods at certain dates (seasonality or maximum extent), but did not explore their dynamics over time, which introduce further research questions relating to threshold stability, automation, and sufficient image availability to reproduce flood dynamics. As Pekel et al., 2016 state "measuring long term changes remains a challenge". Pekel et al.'s work provides a remarkable investigation of water dynamics, however their study did not focus specifically on small water bodies. Yamazaki and Trigg 2016, Mueller et al., 2016 and Yamazaki et al., 2015, who also focussed on global Landsat water inventories, recognise difficulties "due to the presence of both water and vegetation within the pixels" and that their "product may not be fit for [...] small farm dams". Pekel et al. mention omission errors of 23% on seasonal water bodies but "sample pixels of seasonal water bodies within 1° tiles" were used. These therefore do not specifically include or focus on small lakes. Considering the scarcity of ground truth data on small lakes, this is therefore a rare opportunity to specifically quantify the performance on small lakes of long term Landsat monitoring. Ogilvie et al. 2016b focussed on developing an approach for ungauged lakes, where no hypsometric relationships were available and illustrating applications on 51 lakes.

**Pg 3 line 20: the term "low resolution" is a bit confusing. Does it refer to Landsat 30m resolution? Or to medium resolution sensors like the cited MODIS?**

This paragraph refers to the work undertaken to provide long term monitoring at regional and global scale with Landsat imagery. The sentence has therefore been modified to: *"These errors on small lakes are influenced by the insufficient spatial resolution but essentially by the increased presence of flooded vegetation and shallow waters which affect the reflectance signal…"*

**Pg 3 last par: temporality issues can be now better addressed by combining to Sentinel2 (i.e. Kaplan et al. 2017, Du et al. 2016 and Zohu et al. 2017), see general comment above**

This section on page 3 refers to the issues relating to "Landsat imagery to map and monitor small water bodies", as this paper chose to focus on Landsat's ability, considering that "Recent sensors such as Sentinel-2 capture 10 m images of the entire globe every 5 days providing enhanced opportunities but hydrological investigations which require historical perspectives will continue to rely on previous sensors", here Landsat. However as stated, under general comment 1, a specific comparison of the combined benefit of using Landsat and Sentinel-2 since 2015 has now been integrated to the manuscript. The following sentence was added here: "*Finally, the benefits provided by Sentinel-2 imagery for surface water monitoring since 2015 were quantified.*"

**Pg 5 section 2.2. this section should be more clearly written (see point 4 above): how many GPS contours were available? And more important: how in situ areas for the long term analysis were derived? (line 13 refers to water volumes derived from stage values, what about areas?)? An accuracy assessment on the in-situ measurements would be more than welcomed!**

This section has been modified as follows : "*19 GPS contours were acquired, providing a range of flooded surface areas from 0.2 ha to 7.8 ha (Table 2) to test the performance of indices.*"

"*Stage values were converted to surface areas and water volumes using rating curves available for each lake.*" As described in general comment 4, further details and an accuracy assessment of the rating curves has been added to the manuscript.

**Section 2.5: employing different metrics allows a complete evaluation of Landsat performances. However the manuscript is a bit confusing about it (minimum RMSE is used to define the thresholds, PDAI for validation purposes, RMSE and NSE for the long term analysis). A table summarising all the metrics employed would help the reader. For completeness, RMSE could be also reported in Table 2 and mean PDAI in table 3.**

We indeed chose to use a variety of metrics to allow a complete and relevant evaluation of Landsat's performance. Confusion matrices were used as is common practice to quantify the classification accuracy of the water detection indices. PDAI was added to provide a directly relevant assessment of the resulting surface area error on each lake. To calibrate the clouds, shadow and SLC off thresholds (% of interference tolerated) to maximise image availability whilst reducing the detrimental influence of these interferences, RMSE over the 15 years of data were calculated and minimised (using 5% increments of each interference). The skill of the method in terms of flood dynamics and water availability were then calculated to highlight how well the results of the method fit the observed values (NSE) and in terms of the amplitude of the errors (RMSE). RMSE was not included in table 2 as these are errors on single values (i.e. each image and each lake, so only 3 values per lake). Based on the reviewer's recommendation we have gathered in table 2 all metrics relating to the calibration/validation of the water detection indices (i.e. confusion matrix values over the 7 lakes and 3 images). Table 3 presents all the metrics relating to the 7 (different) lakes used in long term monitoring (i.e. results over 7 different lakes and 546 images). Based on the reviewer's recommendation, we have included for completeness PDAI values on surface areas in Table 3, using median PDAI which is less influenced here by individual outliers than mean PDAI.

**Fig 3: this example shows one of the biggest lake analysed. Given the paper focus on small water bodies, it would be interesting to add some examples of MNDWI performances for smaller lakes, and discuss this in term of the amount of vegetated and/or mixed pixels.**

Additional figures of the performance of the MNDWI approach against the DGPS contours on 2 additional small lakes have now been added in Figure 7. Further to our reply under general comment

3, NDVI on SPOT 10m imagery for May 2013 has been used to distinguish open water and vegetated pixels and illustrate here which pixels are correctly classified with MNDWI and with the JRC database.

**Fig. 12 This figure would be more informative if the authors could add the information on the MNDWI points directly derived from Landsat data and those interpolated**

The figure has been modified to add points for the values of the individual Landsat observations.

**Section 3.3.3 lines 21-24: As already pointed out this should be better clarified in the methodology section. An error analysis seems necessary.**

Please see reply under general comments 4 and details in the last paragraph of our reply on the error analysis included.

**Section 3.5 see point 3 in general comments. Fig. 15 Given that the JRC dataset only concern "open water" a 1:1 line should not be expected**

Please also see reply to general comment 3. This figure seeks specifically to show how JRC datasets lead to errors on small water bodies, due to their specificities (which include shallow waters, and standing/floating vegetation) which the method developed by Pekel et al. is currently unable to include. Additional comparisons have now been added to clarify these differences.

**Conclusion pg 23, line 32 reference to SWOT is not appropriate given the focus on small water bodies. SWOT mission spec are indeed given for water bodies with area above 1 km² Line 33: Low cloud: not if radar is used (i.e. Sentinel1)**

The reference to SWOT has been removed. Line 33 has been modified to clarify that these comments refer only to optical sensors and Sentinel-1 has now been specifically mentioned after active sensors.

*"Monitoring flood dynamics with optical sensors remains however dependent on low cloud cover and alternate optimisation of clouds & SLC-off or concomitant imagery sources including active sensors (e.g. Sentinel-1) could also be used to maintain more observations at critical stages such as flood peaks (Eilander et al., 2014)."*

**Appendix A: see point 2 in the general comments concerning the Landsat land surface products**

Please see reply under point 1 in the general comments.

**Technical comments:**

**Fig 7: white dots are difficult to see, please change the color table**

The figure has been modified accordingly.

**Reply to C. Cudennec - Referee #2 - – post revisions to the manuscript**

We wish to thank C. Cudennec again for the valuable and relevant comments provided which have helped strengthen the manuscript. Please find below an updated point by point reply to the issues raised and details of the additions and modifications to the manuscript.

**Ogilvie et al. develop and present an important methodology which values available remote sensing informations and opens low cost applications in the future. It allows assessing and mapping filling of reservoirs spread across a semiarid area, with strong implications for monitoring resources availability across the territory where water is crucial in the Water-Food-Energy-Development nexus; as well as agregation of hydrological impacts on the functioning of the whole basin. The methodology is developed and tested thanks to accurate field campaigns, over a pilot basin in central semiarid Tunisia which is emerging as a strong reference since 15 years. The methodology is based on the use of 7 well known spectral indices; and their assessment thanks to the available Landsat images is well justified and discussed, against field difficulties such as shallow waters, vegetation development, frequency of images regarding rapid dynamic of floods.**

**Litterature review about these 7 indices, and their previous applications and limits could be expanded in section 2.4 as it is too implicit so far. It is refered to a "widely used" status whereas it is mentioned at some places that these are more or less relevant (rural gently sloping watershed, design to detect mixed water reflectance, MNDWI outperformed AWEI when extracting narrow water bodies...).**

This section has been modified to provide additional detail of the applications and specificities of the spectral water indices. The following text has been inserted in the manuscript:

*"Six widely used water detection indices (Table 1) were compared to assess their suitability to monitor floods on small reservoirs in terms of detection accuracy and threshold stability. These include NDWI (Normalised Difference Water Index) developed by McFeeters (1996) exploiting the difference in reflectance of water in the green and NIR bands. Xu (2006) proposed a modified NDWI (MNDWI) where the NIR band was substituted by the SWIR band to improve distinction of built up features over water. In parallel, Lacaux et al. (2007) developed the NDPI (Normalised Difference Pond Index) which also exploits the low reflectance of water in SWIR and its contrast with the green band. NDPI is effectively the opposite of MNDWI and was therefore not investigated separately. Gao (1996) developed a NDWI but using NIR and SWIR bands. Xu (2006) later defined this as NDMI (Normalised Difference Moisture Index). NDTI (Normalised Difference Turbidity Index) was also developed using the red and green bands, and exploits the principle that turbid water reacts like bare soils, i.e. low reflectance in green and high in red. The NDVI (Normalised Difference Vegetation Index) (Rouse et al. 1973) is one of the most well known band ratio index which exploits the contrast between the peak reflectance in the infrared band and the low reflectance in the red to monitor vegetation. It has also been used to detect water bodies (Ma et al. 2007; Mohamed et al. 2004) as the index becomes positive in presence of vegetation and negative for water bodies. Finally, AWEI (Automated Water Extraction Index) (Feyisa et al. 2014) was empirically developed to discriminate water over several large lakes using wavelengths within 5 Landsat bands. Two variants exist, AWEInsh and AWEIsh, the latter being optimised to remove (urban and topographic) shadow pixels. Both can be used in succession in the presence of highly reflective*

*areas (snow, roofs, etc.) but in this rural, semi-arid gently sloping watershed, the AWEInsh was used. For SWIR, band 5 for TM/ETM+ and band 7 for OLI were used in accordance with other studies (Ji et al. 2009; Ouma and Tateishi 2006). For NVDI and NDTI, values below the threshold were classified as water, and conversely for the four other indices."*

**Also a rapid positioning of this approach regarding spatial altimetry and geodesy applied to lakes would be welcome to better assess the challenges of small reservoirs in data scarce regions (See for instance Cretaux et al., 2011, 2015, 2016).**

This relevant comment has led us to add the following text in the introduction of the manuscript.

*"Satellite altimetry originally developed to monitor the ocean's surface has increasingly been exploited and adapted since the 1990s to monitor height variations of continental surface waters (Cretaux et al., 2016). Used across rivers and large lakes, recent works have sought to transpose these to smaller water bodies (from 50 ha) but highlight several limitations relating essentially to the poor density of altimetry tracks, the long along-track path lengths, and low revisit times (Baup et al., 2016, Avisse et al. 2017). Altimeters aboard Topex, Envisat, Jason-1 and Jason-2 have temporal resolutions ranging from 10 to 35 days and high vertical accuracy however their narrow swaths and long along track path lengths (1 km) restrict their application essentially to large lakes (> 100 km², Avisse et al., 2017). Other altimeters have improved spatial resolution and cover more of the globe but at the expense of low revisit times (368 days for Cryostat-2), removing any monitoring possibilities. The Dahiti altimetry database (Schwatke et al., 2015) employed by Busker et al., 2018 combines the tracks of numerous altimeters to optimise the sites covered, temporal resolution and the length of observations. These provide data for 168 sites in Africa, however none in Tunisia and lakes must have a minimum 300 m diameter (circa 7 ha). The Sentinel-3a and 3b constellation provide major improvements in their along track resolution (300 m) making them potentially suitable for lakes around 4 ha but inter-tracks of 52 km mean many lakes are not covered by their trajectories (Cretaux et al., 2016). Furthermore, radar altimetry provides an estimate of the altitude of the water surface, based on the two-way travel time of the radar pulse and the known altitude of the satellite, but assessing absolute volumes requires site specific data such as stage height or topographic data (Baup et al., 2016). Avisse et al., 2017 showed that DEM data taken before lakes were built or when it was empty could be used but on larger lakes (59 ha to 379 ha) with 30 m data."*

Crétaux, J. F., Abarca-del-Río, R., Berge-Nguyen, M., Arsen, A., Drolon, V., Clos, G., & Maisongrande, P. (2016). Lake volume monitoring from space. Surveys in Geophysics, 37(2), 269-305.

Schwatke, C., Dettmering, D., Bosch, W., & Seitz, F. (2015). DAHITI–an innovative approach for estimating water level time series over inland waters using multi-mission satellite altimetry. Hydrology and Earth System Sciences, 19(10), 4345-4364.

Busker, T., de Roo, A., Gelati, E., Schwatke, C., Adamovic, M., Bisselink, B., Pekel, J.-F., and Cottam, A.: A global lake and reservoir volume analysis using a surface water dataset and satellite altimetry, Hydrol. Earth Syst. Sci. Discuss., in review, 2018.

**My major concern is the need to better explicit the hydrometric approach: remote sensing allows to assess water surface. Deducing water storage / volume / availability / resources needs the use of a volume-stage-surface / rating / bathymetric curve. This should be made more explicit in principle in the Introduction and then the Methods section. Further, the bathymetric curves of small reservoirs in this region are not stationnary, as erosion-silting is important, yet very heterogeneous across**

**regions and so reservoirs. The Authors say page 19 that the curve of the biggest reservoir (Mora) is obsolescent because not updated over the past 20 years. This points the need to precisely address the issue of the exact availability and accuracy of bathymetric curves for every considered reservoirs (beyond the short statement on P5, L15 referring to old Albergel and Rejeb, 1997 reference); as well as consequences in terms of uncertainties in the overall method.**

Both reviewers rightly highlighted that further details on the bathymetric curves, their evolution over time from silting and the associated uncertainties are required here. To assess the long-term performance of Landsat imagery to quantify flooded surface areas and volumes, stage data converted using site specific hypsometric (stage-surface-volume) relationships were used.  These were acquired over 1990-2007 through previous research projects and additional levelling of Hoshas as part of this research in 2014.  A figure has been added to the manuscript to explicit the number of relationships for each lake and illustrates the evolution of the rating curves from localised silting over time. Furthermore, accuracy assessments of the hypsometric rating curves and their updates over time have been integrated.  These notably exploit 7 additional GPS contours acquired in 2013 and 2014 on Guettar and Morra. On other lakes, the absence of in situ data to quantify the influence of silting in rating curves until 2014 is overcome based on the silting trends observed on 15 lakes in the region over the 1990-2000 period. These assessments are then used to discuss results and the potential to use Landsat derived surface area estimates at multiple water levels to create and correct the site-specific hypsometric relationships is also illustrated and discussed.

The following text has been inserted in the Methods:

[revised manuscript text omitted]

**Minor issues:**

**-P1, L22: Reservoirs do not reduce soil loss -but sediment transfer once in the network.**

This sentence has been modified as : *"These have been built to reduce sediment transfer and silting of downstream dams, as well as harvest scarce and unreliable water resources for local users (Habi and Morsli, 2011 ; Wisser et al., 2010)"*

**-P2, L20: Okavango and Mekong Deltas.**

The text has been modified accordingly.

**-P3, L33: Localise instead of localised.**

The text has been modified accordingly.

**-Section 2.5: What are the exact dates of the images -and what are the characteristics of the rainfalls over that particular period?**

The precise dates of the images (29.03.2013, 24.05.2013, 09.06.2013) have been included in table 2. The following sentence has also been added: *"Rainfall recorded over this period (March-June 2013) ranged between 27 mm and 44 mm across the lakes concentrated on 1 event on the 24.04.2013."* The gradual decline in water surface area observed on all lakes across these 3 images is coherent with the rainfall characteristics.

**-P9, L17: Provide references about Gouazine basin and reservoir (Nasri et al., for instance).**

Two references have been added here. *"Gouazine lake (Nasri et al., 2004; Al Ali et al., 2008) which possessed both the longest time series (over 15 years) and the most accurate data and rating curves (updated 6 times) was used to optimise the thresholds."*

**-P13, L6: Duplication of "in".**

The text has been modified accordingly.

**-P16, L2: The Merguellil catchment.**

The text has been modified accordingly.

**-P33, L4: author Calvez duplicated.**

The text has been modified accordingly.

[revised manuscript text omitted]